# Vulnerable Data-Aware Adversarial Training

**Yuqi Feng, Jiahao Fan, Yanan Sun**[*]
College of Computer Science, Sichuan University
feng770623@gmail.com, fanjh@scn.edu.cn, ysun@scn.edu.cn

## Abstract

Fast adversarial training (FAT) has been considered as one of the most effective alternatives to the computationally-intensive adversarial training. Generally, FAT methods pay equal attention to each sample of the target task. However, the distance between each sample and the decision boundary is different, learning samples which are far from the decision boundary (i.e., less important to adversarial robustness) brings additional training cost and leads to sub-optimal results. To tackle this issue, we present vulnerable data-aware adversarial training (VDAT) in this study. Specifically, we first propose a margin-based vulnerability calculation method to measure the vulnerability of data samples. Moreover, we propose a vulnerability-aware data filtering method to reduce the training data for adversarial training thus improve the training efficiency. The experiments are conducted in terms of adversarial training and robust neural architecture search on CIFAR-10, CIFAR-100, and ImageNet-1K. The results demonstrate that VDAT is up to 76% more efficient than state-of-the-art FAT methods, while achieving improvements regarding the natural accuracy and adversarial accuracy in both scenarios. Furthermore, the visualizations and ablation studies show the effectiveness of both core components designed in VDAT.

## 1 Introduction

Adversarial training has been demonstrated as one of the most effective techniques [35, 40] to enhance the adversarial robustness of deep neural networks. However, standard adversarial training methods are shown to be time-consuming due to the multi-step gradient calculation and backward propagation for generating the adversarial examples [35, 56]. Regarding this problem, fast adversarial training (FAT) methods [44, 48] are proposed to reduce the computational budget of standard adversarial training. Currently, most of the FAT methods mainly focus on generating adversarial examples more efficiently [39]. In particular, FAT methods often replace the multi-step adversarial example generation with the single-step one, thus improve the efficiency of adversarial training on the whole dataset. However, recent studies demonstrate that some data samples are not important to adversarial robustness, and learning these samples increases the training time but contributes little to the performance [20, 47]. As a result, the efficiency and efficacy of FAT methods is limited.

In order to reduce the scale of training data and further improve FAT methods, some studies have proposed FAT methods based on data filtering [13, 3, 47]. These methods mainly adopt the batch-wise filtering manner to select subsets of the training data. For example, some methods evaluate the contribution of a batch of data based on the related adversarial and natural losses [3], the gradient approximation error of subsets of data [13], or the filling degree of the corresponding input spaces segments [47]. However, these methods do not take the vulnerability of each samples in one batch into account, limiting the performance of adversarial training. Specifically, the distance between different data samples and the decision boundary is different according to previous studies [57, 50].

---

[*]Corresponding author.

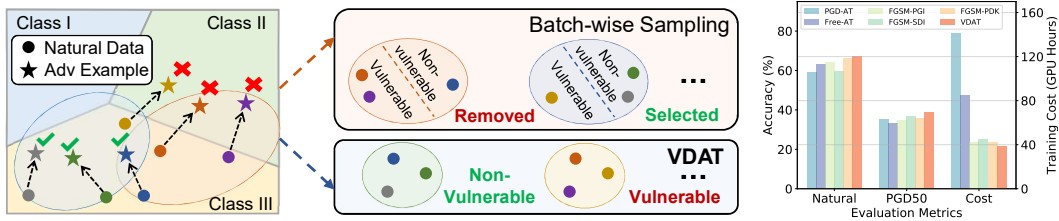

Figure 1: The core idea of VDAT and state-of-the-art FAT methods based on data filtering.

Figure 2: The performance on ImageNet-1K.

Consequently, the vulnerability of data samples is different under the adversarial attack with the same strength. An example is shown in the left part of Figure 1, where the gray, green, and blue samples are less vulnerable than the yellow, red, and purple ones. Because the random allocation of batches, each batch is likely to contain both vulnerable and non-vulnerable samples as shown in the upper right of Figure 1. Suppose the first batch is considered to be less representative than the second batch by the method, and then only the second batch will be learned because of the batch-wise filtering manner. As a result, the vulnerable samples in the first batch are not learned, while the non-vulnerable ones in the second batch are learned, thus hindering the overall performance and efficiency.

In this study, we introduce vulnerable data-aware adversarial training (VDAT) to tackle the above issue of existing FAT methods based on data filtering. As shown in the lower right part of Figure 1, the main idea of VDAT is to perform the sample-wise vulnerability evaluation, and then detect the vulnerable samples and learn them in a targeted way. This goal is achieved by two core components designed. First, we propose a margin-based vulnerability calculation method, so as to determine the vulnerability of each sample in the given dataset. The vulnerability is directly measured by the decision margin between the natural data and adversarial examples, thus the computational cost is low. Second, we propose a vulnerability-aware data filtering strategy, to reduce the number of samples for adversarial training while learn vulnerable samples more robustly. In this strategy, each data sample is given a certain probability to be changed to adversarial example, and the probability is calculated according to the vulnerability of this data sample. The more vulnerable data samples will have larger probability to be changed into adversarial examples. As a result, the vulnerable samples are learned robustly in the training process, thus the overall performance is improved while maintaining low training cost as shown in Figure 2. The contributions of this work are summarized as follows:

- We bring a new perspective to FAT. Different from the mainstream perspective focusing on the batch-wise data filtering, we mainly focus on the sample-wise data filtering to adversarially learn the vulnerable data samples.

- We propose VDAT with the margin-based vulnerability calculation and the vulnerability-aware data filtering, to improve the performance of FAT and lower the computational cost.

- We show that VDAT achieves the state-of-the-art performance regarding both natural and adversarial accuracy on CIFAR-10, CIFAR-100, and ImageNet-1K, while bringing up to 76% efficiency improvement. In addition, VDAT is also effective in improving performance and efficiency in the scenario of robust neural architecture search (NAS).

## 2   Related Work

### 2.1   Fast Adversarial Training (FAT)

FAT is a kind of method designed to accelerate the time-consuming training process of the standard adversarial training [44]. In general, most FAT methods mainly focus on the efficient generation of adversarial examples [1, 25]. For example, Jia *et al.* [28] propose a input data-aware adversarial example initialization method, to generate adversarial examples which are beneficial to both natural accuracy and adversarial robustness in a single-step manner. Tong *et al.* [45] propose the taxonomy driven FAT method to overcome the catastrophic overfitting [30] via optimizing learning objective, loss function, and initialization manner jointly. Jia *et al.* [27] propose prior knowledge-guided

adversarial example initialization method, to improve the quality of adversarial examples generated and overcome the catastrophic overfitting. Although these methods have achieved decent performance, they still need to train on the whole dataset with samples contributing little to the adversarial robustness [47], thus limiting the efficiency. Therefore, we mainly focus on the perspective about data filtering in this study, to further improve the efficiency of FAT.

## 2.2 Decision Boundary Analysis

The studies regarding decision boundary analysis mainly lie in three perspectives, i.e., enhancing the adversarial training, improving the adversarial attacks, and interpreting the decision mechanism of deep neural networks. In the first category, existing methods mainly focus on dynamically adjusting the decision boundary during the training [11, 42, 50, 4], to improve both natural accuracy and adversarial robustness of the model. In the second category, existing studies leverage the decision boundary information to enhance both white-box and black-box attacks based on learning [9, 43], to evaluate the adversarial robustness more accurately. In the third category, the mainstream direction is to explore the relationship between the data, features, and the decision boundary during the adversarial training [16, 37]. Then, the information about the features and decision boundary can be adopted to interpret the shift of decision boundary of the neural network during adversarial training. Different from these studies, we focus on data filtering from the decision boundary perspective in this work. Specifically, we estimate the vulnerability of data samples to determine vulnerable ones based on decision boundary analysis, to improve both efficiency and efficacy of FAT.

## 2.3 Robust Neural Architecture Search (NAS)

Robust NAS aims to automatically design robust neural architectures against adversarial attacks [24, 14, 18]. Generally, most robust NAS methods are constructed based on differentiable NAS methods [34, 5, 51], which need to optimize the trainable weights and architectures parameters in a supernet. To ensure the adversarial robustness of the derived architectures, robust NAS methods often adopt adversarial training techniques in the optimization of above variables. For instance, Mok *et al.* [36] adopt adversarial training to obtain the input-loss landscape, and then update the architecture parameters based on this indicator. Qian *et al.* [41] adopt adversarial training to update trainable weights and architecture parameters, to reduce the non-robust feature distortion and enhance the adversarial robustness of derived architectures. Ou *et al.* [38] propose a multi-objective search strategy to balance the losses of natural training and adversarial training. In our experiments, we replace the adversarial training in robust NAS with VDAT and other state-of-the-art FAT methods, to evaluate the scalability of VDAT in terms of searching for adversarially robust neural architectures.

# 3 Preliminaries

According to the common practice of adversarial training [27], we define the target neural network parameterized by $\theta$ as $f_\theta$: $\mathbb{R}^d \to \mathbb{R}$. Given a data sample $x \in \mathcal{X}$ and the corresponding label $y \in \{1, 2, \ldots, k\}$, the network can be further denoted as $f_\theta(x) = \arg\max_{c \in \{1,2,\ldots,k\}} \phi_\theta^c(x)$, where $\phi_\theta^c(x)$ denotes the logit for class $c$. Based on this, the decision margin $\mathcal{M}_\theta^y(x)$ of the data sample $x$ can be denoted as Equation (1) according to the previous study [50].

$$\mathcal{M}_\theta^y(x) = \phi_\theta^y(x) - \max_{y' \neq y} \phi_\theta^{y'}(x) \tag{1}$$

The main goal of the proposed VDAT method is to detect the vulnerable data samples, i.e., data samples having relatively smaller decision margin between themselves and the corresponding adversarial examples. Then, the data samples detected are collected to construct a subset of data for adversarial training, thus reducing the computational budget and improving the overall performance.

# 4 Vulnerable Data-Aware Adversarial Training

The overall framework of VDAT is shown in Figure 3. As can be seen, the proposed VDAT method works on the natural data samples, through adversarial perturbation, and two core components, i.e., the margin-based vulnerability calculation and the vulnerability-aware data filtering, then the data samples for natural training and adversarial training are determined and adopted to train the neural network. The details of both core components are presented in the following sections.

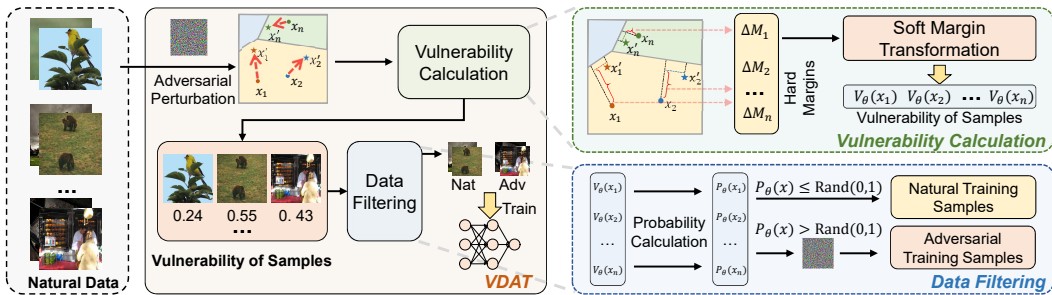

Figure 3: The illustration about the overall framework of VDAT. After adding adversarial perturbations to generate adversarial examples, the margin-based vulnerability calculation and the vulnerability-aware data filtering are performed sequentially.

## 4.1 Margin-based Vulnerability Calculation

After generating adversarial examples $\mathcal{X}'$ in the first step of VDAT, the margin-based vulnerability calculation will play the role in calculating the vulnerability of each sample in $\mathcal{X}$. Specifically, the vulnerability $\mathcal{V}_\theta(x_i)$ of $x_i$ is represented by the difference between the logit margins of both natural sample $x_i \in \mathcal{X}$ and the corresponding adversarial example $x_i' \in \mathcal{X}'$, as shown in Equation (2):

$$\mathcal{V}_\theta(x_i) = - \mid \mathcal{M}_\theta^y(x_i) - \mathcal{M}_\theta^y(x_i') \mid \tag{2}$$

where $y$ denotes the true label of the data sample $x_i$. Besides, the physical significance of the subtraction is the margin difference in the decision space between $x_i$ and $x_i'$. The larger absolute value of that subtraction denotes the larger margin difference, indicating the natural data is harder to be attacked and has lower vulnerability.

However, both $\mathcal{M}_\theta^y(x_i)$ and $\mathcal{M}_\theta^y(x_i')$ are hard margins formulated by the maximum operator. In this way, the logit margin is calculated only based on logits of the class $y$ and the class with the largest logit given by the model $f_\theta$. Unfortunately, there existing targeted adversarial attacks [35], and the adversarial example $x_i$ can be misclassified into each possible class based on such kind of attacks. Therefore, formulating the logit margin with only one class $y' \neq y$ cannot well cover all possible situations. To this end, we formulate the soft margin $\mathcal{S}_\theta^y(x_i, \tau)$ and $\mathcal{S}_\theta^y(x_i', \tau)$ for both $x_i$ and $x_i'$ motivated by [50]. The detailed formulation is shown in Equations (3) and (4), where $\tau$ is a pre-determined hyperparameter.

$$\mathcal{S}_\theta^y(x_i) = \phi_\theta^y(x_i) - \frac{1}{\tau} \log \sum_{y' \neq y} \exp(\tau \phi_\theta^{y'}(x_i)) \tag{3}$$

$$\mathcal{S}_\theta^y(x_i') = \phi_\theta^y(x_i') - \frac{1}{\tau} \log \sum_{y' \neq y} \exp(\tau \phi_\theta^{y'}(x_i')) \tag{4}$$

Based on the soft margins, the vulnerability of the data sample $x_i$ can be rewritten as Equation (5).

$$\mathcal{V}_\theta(x_i) = - \mid \mathcal{S}_\theta^y(x_i) - \mathcal{S}_\theta^y(x_i') \mid \tag{5}$$

In the following, we discuss the motivation of adopting the margin difference instead of the pure margin of the single data sample. The reason is that the pure margin only reflects the output situation of the model and cannot represent the information of decision boundary [50]. In contrast, adopting the margin difference can avoid the above problem, this is because the pure margin of adversarial examples carries the decision boundary information [29]. Specifically, the decision boundary [33] is characterized based on a certain data sample $\hat{x}$, the true label $\hat{y}$ of $\hat{x}$, and another label $y' \neq \hat{y}$. If this data sample satisfies $\phi_\theta^{\hat{y}}(\hat{x}) = \phi_\theta^{y'}(\hat{x})$, then we assume that the data sample $\hat{x}$ is on the decision boundary between classes $\hat{y}$ and $y'$. In practice, the problem is that the sample $\hat{x}$ often cannot be accurately found [50]. Fortunately, previous studies [9] state that the adversarial attack can push the sample towards the decision boundary, thus the logit margin of adversarial examples can well reflect the decision boundary information in some cases. Based on this, we adopt the margin difference to calculate the vulnerability in this study.

## 4.2 Vulnerability-Aware Data Filtering

Based on the vulnerability $\mathcal{V} = \{\mathcal{V}_\theta(x_1), \mathcal{V}_\theta(x_2), \ldots, \mathcal{V}_\theta(x_n)\}$ calculated for the natural data samples $\{x_1, x_2, \ldots, x_n\}$, the data samples are filtered for natural training and adversarial training accordingly. Specifically, each sample $x_i$ is given a probability $\mathcal{P}_\theta(x_i)$ to be changed to the adversarial example for adversarial training. The higher the vulnerability of $x_i$ is, the larger the probability $\mathcal{P}_\theta(x_i)$ is, in order to help the model learn this vulnerable sample more robustly. The details of the calculation of $\mathcal{P}_\theta(x_i)$ are presented as follows.

To build the connection between the vulnerability and the probability $\mathcal{P}_\theta(x_i)$, we normalized the vulnerability $V_\theta(x_i)$, for the convenience of determining the probability. In particular, the calculation for the probability $\mathcal{P}_\theta(x_i)$ is formulated as Equation (6):

$$\mathcal{P}_\theta(x_i) = \frac{1}{2} \left( \frac{\mathcal{V}_\theta(x_i) - \overline{\mathcal{V}}_\theta(x_1, x_2, \ldots, x_n)}{\max\{|\ \mathcal{V}_\theta(x_i) - \mathcal{V}_\theta(x_k)\ |\}_{k=1,2,\ldots,n}} + 1 \right) \tag{6}$$

where $\overline{\mathcal{V}}_\theta(x_1, x_2, \ldots, x_n)$ denotes the average value of the vulnerability in $\mathcal{V}$. The range of $\mathcal{P}_\theta(x_i)$ is narrowed from zero to one by calculating in this way.

After determining the probability for each data sample, the data filtering can be performed accordingly. Specifically, the filtering process is described by Equation (7):

$$\mathcal{F}(x_i) = \begin{cases} x_i + \delta, & \mathcal{R}(0,1) \leq \mathcal{P}_\theta(x_i) \\ x_i, & \mathcal{R}(0,1) > \mathcal{P}_\theta(x_i) \end{cases} \tag{7}$$

where $\delta$ denotes the adversarial noise generated by the specific adversarial attack of adversarial training, $\mathcal{R}(0,1)$ denotes a uniform random value ranged from zero to one. In this way, the data samples with the higher vulnerability are more likely changed to adversarial examples, to help the target model learn them more robustly.

When the filtered data $\mathcal{F}(x_i)$ is determined for all data samples in $\mathcal{X} = \{x_1, x_2, \ldots, x_n\}$, the datasets for natural training and adversarial training can be obtained. Supposing there are $m$ data samples filtered for adversarial training, we denote the datasets for natural and adversarial training as $\mathcal{X}_{\text{adv}} = \{x_1, x_2, \ldots, x_m\}$ and $\mathcal{X}_{\text{nat}} = \{x_{m+1}, x_{m+2}, \ldots, x_n\}$, then the overall training loss $\mathcal{L}_{\text{train}}$ is formulated as Equation (8).

$$\mathcal{L}_{\text{train}} = \mathcal{L}_{\text{nat}}(\mathcal{X}_{\text{nat}}, \theta) + \mathcal{L}_{\text{adv}}(\mathcal{X}_{\text{adv}}, \theta) \tag{8}$$

Based on this, we present the whole training process of VDAT in Algorithm 1. As can be seen, the data filtering is performed with a predefined interval $\mathcal{T}$ in the whole training process (line 2). During the data filtering, the probability $\mathcal{P}_\theta(x_i)$ is calculated in the first step (line 3), and then the data filtering and the update of $\mathcal{X}_{\text{nat}}$ and $\mathcal{X}_{\text{adv}}$ are performed sequentially (lines 4 and 5). Moreover, the network parameter $\theta$ is updated based on calculating the training loss $\mathcal{L}_{\text{train}}$ in each epoch (line 7). Finally, the trained model can be obtained. It is worth noting that we introduce a hyperparameter $\mathcal{T}$ indicating the interval of data filtering. This is because the data filtering introduces additional computational budget

---

**Algorithm 1** VDAT

**Input:** Target dataset $\mathcal{X}$, target model with parameter $\theta$, training epoch $\mathcal{N}$, and interval $\mathcal{T}$.
**Output:** The trained model with parameter $\theta^*$.
1: **for** $i = 0$ to $\mathcal{N}$ **do**
2:     **if** $i \mod \mathcal{T} == 0$ **then**
3:         Calculate $\mathcal{P}_\theta(x_i)$ for each sample in $\mathcal{X}$
4:         Perform data filtering in Equation (7)
5:         Update datasets $\mathcal{X}_{\text{nat}}$ and $\mathcal{X}_{\text{adv}}$
6:     **end if**
7:     Calculate $\mathcal{L}_{\text{train}}$ and update $\theta$
8: **end for**
9: **Return** The trained model with $\theta^*$

---

in the training process due to the vulnerability calculation, and this hyperparameter can balance the total training cost and the performance of the trained model. Detailed analysis of this hyperparameter is presented in our experiments. Besides, additional details regarding the workflow of VDAT can be found in **Appendix A**.

## 4.3 Time Complexity Analysis

In this section, we present the time complexity analysis of the proposed VDAT method. For the margin-based vulnerability calculation, because we need to iterate over all $n$ samples in the target

dataset $\mathcal{X}$, the time complexity for this process is $\mathcal{O}(n)$. Meanwhile, calculating the soft margin for each sample needs to iterate the logits for all $k$ classes. As a result, the time complexity for this process is $\mathcal{O}(k)$, and the overall time complexity for the margin-based vulnerability calculation is $\mathcal{O}(nk)$. For the vulnerability-aware data filtering, we also need to iterate all $n$ data samples and calculate the probability $\mathcal{P}$ of each data sample, and the time complexity for this process is $\mathcal{O}(n)$. In summary, because the above two parts are performed serially, the overall time complexity is $\mathcal{O}(nk)$.

## 5 Experiments

### 5.1 Experimental Settings

In order to comprehensively demonstrate the effectiveness of VDAT, we adopt VDAT to train baseline models, i.e., ResNet [23] and WideResNet [54], which are widely used in the community of adversarial training to test the performance of adversarial training techniques [8, 27]. Besides, the benchmark datasets for adversarial training or robust NAS are CIFAR-10, CIFAR-100 [31], and ImageNet-1K [12], while the adversarial attacks for evaluations are FGSM [21], PGD [35], C&W [2], and AutoAttack (AA) [10]. These choices also follow the conventions in the communities of adversarial training and robust NAS [50, 27, 19]. Besides, the hyperparameter $\tau$ of VDAT is set to five and the interval $\mathcal{T}$ is set to ten. The adversarial perturbation adopted before the vulnerability calculation is generated by the FGSM attack. Moreover, because VDAT can be combined with different types of adversarial perturbations for adversarial training, we adopt commonly used FGSM and PGD adversarial perturbations. These two variants of VDAT are denoted as VDAT$_{\text{FGSM}}$ and VDAT$_{\text{PGD}}$ in our experiments, respectively. More details regarding the experimental settings are in **Appendix B**.

Table 1: Experimental results of the adversarial training on CIFAR-10 and CIFAR-100. The baseline models are ResNet18 and WideResNet34-10. The best results are in **bold** and the second best results are underlined.

| Models | Methods | Natural (%) | Adversarial Attacks | | | | | Training Cost |
|---|---|---|---|---|---|---|---|---|
| | | | FGSM (%) | PGD$^{20}$ (%) | PGD$^{50}$ (%) | C&W (%) | AA (%) | (GPU Hours) |
| **CIFAR-10** | | | | | | | | |
| ResNet18 | PGD-AT [35] | 82.32 | 55.12 | 52.83 | 52.60 | 51.08 | 48.68 | 4.45 |
| | FGSM-SDI [28] | 84.86 | 59.96 | 52.54 | 52.18 | 51.00 | 48.50 | 1.38 |
| | TDAT [45] | 82.67 | 55.82 | 49.31 | 49.02 | 50.37 | 46.47 | 2.29 |
| | FGSM-PGK [27] | 81.58 | 62.04 | 55.51 | 55.31 | 51.17 | 49.51 | 1.30 |
| | **VDAT$_{\text{PGD}}$** | 82.96 | 62.26 | 56.08 | 56.07 | **52.85** | **53.89** | 1.96 |
| | **VDAT$_{\text{FGSM}}$** | **86.32** | **62.94** | **56.30** | **56.23** | 51.31 | 53.23 | **1.04** |
| WideResNet34-10 | PGD-AT [35] | 85.17 | 59.78 | 55.07 | 54.87 | 53.84 | 51.67 | 31.90 |
| | FGSM-SDI [28] | 86.40 | 60.68 | 54.95 | 54.60 | 53.68 | 51.17 | 9.40 |
| | TDAT [45] | 87.69 | 60.62 | 43.38 | 41.53 | 46.71 | 22.58 | 9.09 |
| | FGSM-PGK [27] | 83.32 | 63.26 | 58.23 | 57.46 | 54.33 | 53.28 | 8.30 |
| | **VDAT$_{\text{PGD}}$** | 88.10 | 64.40 | **60.36** | **60.26** | **55.70** | **54.99** | 9.77 |
| | **VDAT$_{\text{FGSM}}$** | **92.34** | **65.08** | 59.49 | 58.30 | 55.42 | 54.49 | **3.41** |
| **CIFAR-100** | | | | | | | | |
| ResNet18 | PGD-AT [35] | 57.50 | 30.95 | 29.00 | 28.90 | 27.60 | 25.48 | 4.73 |
| | FGSM-SDI [28] | 58.54 | 37.19 | 27.99 | 27.67 | 25.85 | 23.27 | 1.65 |
| | TDAT [45] | 57.32 | 40.29 | 33.17 | 33.06 | 28.47 | 26.61 | 2.29 |
| | FGSM-PGK [27] | 56.27 | 37.40 | 32.85 | 29.92 | 28.39 | 26.86 | 1.30 |
| | **VDAT$_{\text{PGD}}$** | 54.44 | 40.36 | 34.29 | 34.26 | **29.34** | 31.44 | 2.01 |
| | **VDAT$_{\text{FGSM}}$** | **59.43** | **41.17** | **35.69** | **35.61** | 28.23 | **32.39** | **1.05** |
| WideResNet34-10 | PGD-AT [35] | 60.63 | 31.39 | 26.21 | 25.92 | 23.96 | 21.41 | 31.90 |
| | FGSM-SDI [28] | 63.49 | 40.94 | 30.80 | 25.58 | 13.71 | 22.29 | 9.40 |
| | TDAT [45] | 63.07 | 40.51 | 29.04 | 27.77 | 29.16 | 25.15 | 9.25 |
| | FGSM-PGK [27] | 64.22 | 40.54 | 30.04 | 29.73 | 27.82 | 26.11 | 8.30 |
| | **VDAT$_{\text{PGD}}$** | 64.50 | 41.88 | **35.28** | **35.18** | **33.97** | **32.61** | 9.92 |
| | **VDAT$_{\text{FGSM}}$** | **72.91** | **41.90** | 34.64 | 33.53 | 31.01 | 31.65 | **3.44** |

### 5.2 Overall Results for Adversarial Training

To demonstrate the efficiency and efficacy of VDAT, we compare VDAT with state-of-the-art FAT methods focusing on 1) efficient generation of adversarial examples and 2) data filtering. Specifically, we adopt different adversarial training methods to train baseline models, and then the natural accuracy, adversarial accuracy under different adversarial attacks, and the training cost are reported. The benchmark datasets for this set of experiments are CIFAR-10, CIFAR-100, and ImageNet-1K.

As for the first category of FAT methods, the experimental results on CIFAR-10 and CIFAR-100 are presented in Table 1. As can be seen, both VDAT$_{FGSM}$ and VDAT$_{PGD}$ achieve state-of-the-art natural accuracy and adversarial accuracy in most cases. Meanwhile, because VDAT only performs adversarial training on the filtered subset, VDAT$_{FGSM}$ is more efficient than the state-of-the-art FGSM-based FAT methods, i.e., FGSM-SDI, TDAT, and FGSM-PGK. Moreover, VDAT$_{PGD}$ is also much more efficient than standard PGD-AT, and it does not increase much training cost compared with FGSM-based FAT methods. Furthermore, the experimental results in Table 1 also show that VDAT$_{FGSM}$ performs better than VDAT$_{PGD}$ on natural accuracy and weak adversarial attacks, while VDAT$_{FGSM}$ is not as good as VDAT$_{PGD}$ on stronger attacks. Specifically, VDAT$_{FGSM}$ outperforms VDAT$_{PGD}$ in terms of natural accuracy and adversarial accuracy under the FGSM adversarial attack. As for the stronger adversarial attacks, i.e., C&W and AA, VDAT$_{PGD}$ often performs better than VDAT$_{FGSM}$. This is mainly because VDAT$_{PGD}$ provides stronger adversarial examples, and learning these adversarial examples can help the model achieve better accuracy under stronger adversarial attacks. However, it is shown that stronger adversarial examples can lead to robust overfitting of the strong adversarial perturbations [53]. As a result, the model suffers from the poor generalizability to the clean data and weak adversarial examples, thus the natural accuracy and adversarial accuracy under weak adversarial attacks become lower.

Beyond CIFAR-10 and CIFAR-100, we also conduct experiments on ImageNet-1K against state-of-the-art FAT methods in the first category, to demonstrate effectiveness and efficiency of VDAT on the large-scale dataset. As can be observed from Table 2, VDAT achieves the highest natural accuracy and adversarial accuracy under both PGD$^{10}$ and PGD$^{50}$ adversarial attacks. Meanwhile, VDAT is at least 9.6% more efficient than the state-of-the-art FGSM-based FAT methods. In conclusion, VDAT demonstrates superior efficiency and efficacy compared with state-of-the-art FAT methods in the first category.

Table 2: Experimental results of the adversarial training on ImageNet-1K. The baseline model for training is ResNet50.

| Methods | Natural (%) | Adversarial Attacks | | Training Cost |
| | | PGD$^{10}$ (%) | PGD$^{50}$ (%) | (GPU Hours) |
| --- | --- | --- | --- | --- |
| PGD-AT [35] | 59.19 | 35.87 | 35.41 | 140.80 |
| Free-AT [44] | 63.42 | 33.22 | 33.08 | 85.13 |
| FGSM-PGI [26] | 64.32 | 36.24 | 34.93 | _42.47_ |
| FGSM-SDI [28] | 59.62 | 37.50 | 36.63 | 44.53 |
| FGSM-PGK [27] | 66.24 | 37.13 | 35.70 | _42.47_ |
| **VDAT$_{FGSM}$** | **67.47** | **39.48** | **38.69** | **38.41** |

Table 3: Comparisons between the state-of-the-art data filtering-based FAT and VDAT on CIFAR-10, CIFAR-100, and ImageNet-1K.

| Models / Datasets | Methods | Natural (%) | AA (%) | Cost |
| --- | --- | --- | --- | --- |
| WideResNet34-10 / CIFAR-10 | AdvGradMatch [13] | 84.54 | 47.83 | 17.21 |
| | DFEAT [3] | 85.66 | 52.77 | 10.30 |
| | **VDAT$_{FGSM}$** | **92.34** | **54.49** | **3.41** |
| WideResNet34-10 / CIFAR-100 | DFEAT [3] | 66.51 | 30.68 | 14.33 |
| | **VDAT$_{FGSM}$** | **72.91** | **31.65** | **3.44** |
| ResNet50 / ImageNet-1K | DFEAT [3] | 64.76 | 36.10 | 53.72 |
| | **VDAT$_{FGSM}$** | **67.47** | **36.98** | **38.41** |

Regarding the FAT methods falling into the second category, we choose the state-of-the-art methods, i.e., AdvGradMatch [13] and DFEAT [3], and then directly report the experimental results presented in their seminal papers. The experimental results are shown in Table 3. As can be seen, VDAT demonstrates superior performance on all three datasets comparing with the FAT methods selected. Meanwhile, the training cost of VDAT is significantly lower ($\geq 28.5\%$ lower) than that of AdvGrad-Match and DFEAT on both medium scale datasets (CIFAR-10 and CIFAR-100) and the large-scale dataset (ImageNet-1K), thus illustrating the superiority of VDAT.

## 5.3 Overall Results for Robust NAS

In order to evaluate the scalability of VDAT in different scenarios, we conduct experiments in terms of robust NAS. Specifically, we directly replace the PGD-based adversarial training with different FAT methods in the optimization of the supernet, to obtain robust neural architectures. Then, the derived architectures are adversarially trained by the PGD$^{7}$ adversarial training following the conventions [38, 17]. The experimental results are shown in Table 4.

As can be seen from Table 4, the natural accuracy becomes higher when the robust NAS method is combined with VDAT, and this accuracy is also higher than that in cases of FGSM-SDI and FGSM-PDK. Meanwhile, the adversarial accuracy reaches the best in most cases after introducing VDAT$_{PGD}$ or VDAT$_{FGSM}$. Besides, the search cost becomes the lowest after introducing VDAT$_{FGSM}$, and the search cost also becomes lower than that of vanilla robust NAS methods after introducing VDAT$_{PGD}$. These experiments demonstrate the scalability of VDAT in the scenario of robust NAS. Furthermore, it can be observed that VDAT$_{FGSM}$ generally performs better than VDAT$_{PGD}$ on both natural accuracy and adversarial accuracy. This phenomenon is mainly caused by the overfitting phenomenon of the

Table 4: Experimental results of VDAT on CIFAR-10 under white-box attacks. The adversarial training methods are integrated into the corresponding robust NAS methods.

| Models | Params | Natural (%) | Adversarial Attacks | | | | | Search Cost |
| | | | FGSM (%) | PGD$^{20}$ (%) | PGD$^{100}$ (%) | C&W (%) | AA (%) | (GPU Days) |
| --- | --- | --- | --- | --- | --- | --- | --- | --- |
| DSRNA [24] | 2.0M | 80.93 | 54.49 | 49.11 | 48.89 | 38.92 | 44.87 | 0.80 |
| + FGSM-SDI [28] | 1.9M | 86.32 | 55.38 | 43.62 | 43.04 | 16.17 | 41.64 | 0.53 |
| + FGSM-PGK [27] | 2.7M | 85.98 | 59.40 | 50.65 | 50.13 | 52.64 | 48.05 | 0.43 |
| **+ VDAT**$_{PGD}$ | 3.2M | 87.21 | 62.24 | 53.29 | 53.13 | 52.98 | 48.16 | 0.65 |
| **+ VDAT**$_{FGSM}$ | 3.4M | **91.10** | **84.23** | **64.01** | **54.64** | **55.89** | **50.26** | **0.34** |
| AdvRush [36] | 4.2M | 87.30 | 60.87 | 53.07 | 52.80 | 45.13 | 50.05 | 1.22 |
| + FGSM-SDI [28] | 3.2M | 86.42 | 59.66 | 50.78 | 50.24 | 28.06 | 47.89 | 0.74 |
| + FGSM-PGK [27] | 3.7M | 86.28 | 59.26 | 50.31 | 49.74 | 51.86 | 47.15 | 0.59 |
| **+ VDAT**$_{PGD}$ | 3.5M | 87.89 | 62.60 | 54.27 | 54.11 | **52.59** | 52.25 | 0.76 |
| **+ VDAT**$_{FGSM}$ | 3.6M | **88.05** | **63.35** | **55.32** | **54.84** | 46.42 | **54.38** | **0.47** |
| RNAS [41] | 3.5M | 84.13 | 61.90 | 53.48 | 53.35 | 50.74 | 50.54 | 0.71 |
| + FGSM-SDI [28] | 4.3M | 84.55 | 58.32 | 49.38 | 48.97 | 54.08 | 45.65 | 0.42 |
| + FGSM-PGK [27] | 3.3M | 84.68 | 55.24 | 45.47 | 44.89 | 49.89 | 43.84 | 0.34 |
| **+ VDAT**$_{PGD}$ | 4.1M | **87.24** | 62.23 | 54.09 | 53.96 | **54.70** | 51.86 | 0.68 |
| **+ VDAT**$_{FGSM}$ | 3.7M | 87.18 | **62.99** | **55.43** | **55.13** | 54.61 | **53.54** | **0.27** |
| ARNAS [38] | 4.5M | 85.92 | 62.45 | 55.87 | 55.43 | 28.14 | 52.66 | 2.95 |
| + FGSM-SDI [28] | 5.0M | 86.86 | 60.51 | 51.59 | 50.92 | 26.25 | 48.33 | 1.81 |
| + FGSM-PGK [27] | 3.7M | 87.11 | 60.74 | 52.92 | 52.45 | 25.31 | 50.43 | 1.44 |
| **+ VDAT**$_{PGD}$ | 5.7M | 88.88 | 63.88 | 56.85 | 56.43 | **38.24** | 54.73 | 2.22 |
| **+ VDAT**$_{FGSM}$ | 5.0M | **89.34** | **65.67** | **57.15** | **56.81** | 30.34 | **56.11** | **1.15** |

supernet. Specifically, VDAT$_{PGD}$ will cause robust overfitting more easily than VDAT$_{FGSM}$ during the training process of the supernet [53]. As indicated in the previous study [55], the overfitting can make the NAS algorithm add more parameter-free operations (e.g., skip connection) to the derived architecture. As a result, the architectures searched with VDAT$_{PGD}$ demonstrate lower natural accuracy and adversarial accuracy than those searched with VDAT$_{FGSM}$. Please note that additional experimental results regarding robust NAS can be found in **Appendix C**.

## 5.4 Visualizations of VDAT

**Visualizations of the Training Process.** Because the data filtering in VDAT changes the training data for both natural training and adversarial training iteratively, we explore the convergence of VDAT by visualizing the training process, in order to show the reliability of VDAT. As can be seen from Figure 4, no matter which dataset or baseline model is adopted, the training of VDAT converges well. Please note that the significant accuracy improvement at the 100-th training epoch is mainly caused by the decay of the learning rate. This practice follows the convention of adversarial training [28]. Besides, because the data filtering method can change the training data iteratively, VDAT does not suffer from the catastrophic overfitting [30] even though the FGSM-based adversarial training is combined with VDAT.

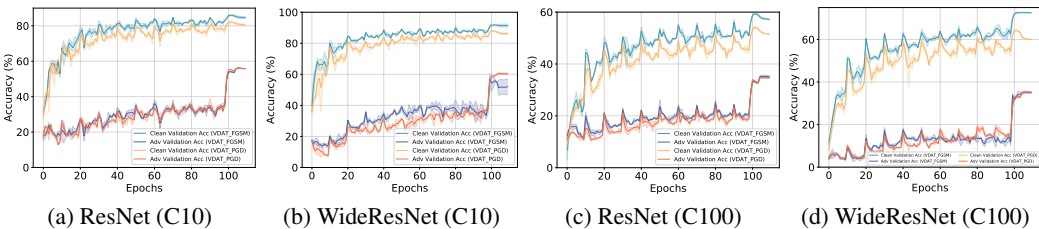

|   (a) ResNet (C10)   |   (b) WideResNet (C10)   |   (c) ResNet (C100)   |   (d) WideResNet (C100)   |

Figure 4: Visualizations of the training processes of VDAT$_{FGSM}$ and VDAT$_{PGD}$ on CIFAR-10 (C10) and CIFAR-100 (C100).

**Visualizations of the Vulnerability of Data Samples.** To further show the effectiveness of VDAT regarding learning vulnerable samples, we visualize the normalized margin difference before and after training. A larger normalized margin difference indicates lower vulnerability. As shown in Figure 5, the distribution shifts to right after training with VDAT on all three datasets, demonstrating VDAT is indeed effective in learning vulnerable data samples robustly. Additional visualizations for the training process and the normalized margin difference can be found in **Appendix D**.

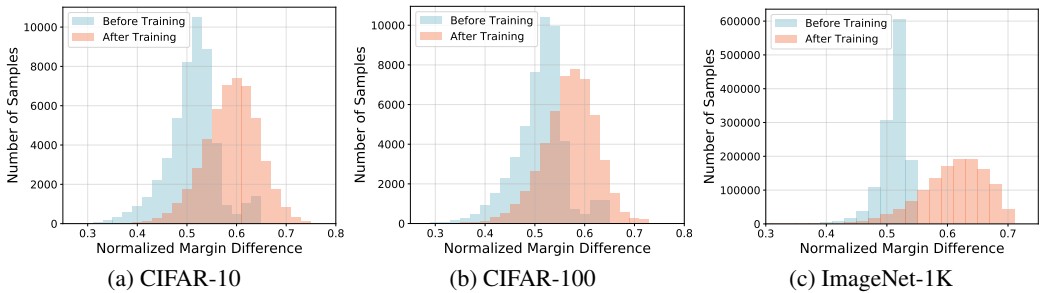

(a) CIFAR-10    (b) CIFAR-100    (c) ImageNet-1K

Figure 5: Visualizations of the normalized margin difference of data before and after training by VDAT$_{\text{FGSM}}$. The baseline models are ResNet18 (CIFAR-10/100) and ResNet50 (ImageNet-1K).

## 5.5    Parameter Studies and Ablation Studies

**Parameter Studies.** We conduct parameter studies for the hyperparameter $\tau$ and the interval $\mathcal{T}$. As shown in Table 5, the performance raises when $\tau$ increases from 0.05 to five, and then slightly decreases when $\tau$ increases from five to 500, illustrating the validity of setting $\tau$ to five. Moreover, as shown in Table 6, both natural accuracy and adversarial accuracy decrease when $\mathcal{T}$ becomes larger, but the training cost becomes lower. To balance the accuracy and training cost, we choose to set $\mathcal{T}$ to 10 as default.

Table 5: Parameter study for the hyperparameter $\tau$ on CIFAR-10. The baseline model for adversarial training is ResNet18.

| $\tau$ | Natural (%) | PGD$^{20}$ (%) | PGD$^{50}$ (%) |
|---|---|---|---|
| 0.05 ($\times$0.01) | 85.88 | 48.53 | 48.41 |
| 0.5 ($\times$0.1) | 85.96 | 53.95 | 53.90 |
| **5 (Default)** | **86.32** | **56.30** | **56.23** |
| 50 ($\times$10) | 86.25 | 54.14 | 54.00 |
| 500 ($\times$100) | **86.32** | 53.27 | 53.12 |

Table 6: Parameter study for the interval $\mathcal{T}$ on CIFAR-10. The training cost is measured in GPU hours.

| $\mathcal{T}$ | Natural (%) | PGD$^{50}$ (%) | Cost |
|---|---|---|---|
| 2 ($\times$0.2) | **91.37** | **58.57** | 1.47 |
| 5 ($\times$0.5) | 88.39 | 54.74 | 1.20 |
| **10 (Default)** | 86.32 | 56.23 | 1.04 |
| 20 ($\times$2) | 82.94 | 44.95 | 0.69 |
| 50 ($\times$5) | 81.81 | 43.91 | **0.63** |

**Ablation Studies.** The ablation studies mainly focus on the filtering method, the kinds of adversarial perturbation adopted before the vulnerability calculation, the soft margin along with the temperature parameter $\tau$ adopted, and the different adversarial training frameworks.

As can be seen in Table 7, when our filtering method is removed or replaced by the random filtering, both natural and adversarial accuracy drops, demonstrating the effectiveness of the filtering method designed. Furthermore, as shown in Table 8, the natural accuracy and adversarial accuracy do not show significant changes when changing the types of adversarial attacks, but the search cost increases when changing FGSM to PGD. Therefore, we choose FGSM attack as default.

Table 7: Ablation study for the filtering method on CIFAR-10. The baseline model for training is ResNet18.

| Filtering? | Natural (%) | PGD$^{20}$ (%) | PGD$^{50}$ (%) |
|---|---|---|---|
| Without | 73.81 | 41.55 | 41.26 |
| Random | 85.96 | 50.96 | 48.84 |
| **VDAT** | **86.32** | **56.30** | **56.23** |

Table 8: Ablation study for the kind of adversarial perturbation adopted before the vulnerability calculation.

| Perturbation | Natural (%) | PGD$^{50}$ (%) | Cost |
|---|---|---|---|
| PGD$^{10}$ | 85.96 | 58.03 | 1.10 |
| PGD$^{20}$ | 85.96 | **58.50** | 1.34 |
| FGSM | **86.32** | 56.23 | **1.04** |

Moreover, we conduct ablation studies regarding the soft margin along with the temperature parameter $\tau$. The effectiveness of the above design is evaluated under the targeted attacks. Specifically, we train ResNet18 on CIFAR-10 with two types of margins, i.e., the pure margin in Equation (2) and the soft margin with different values of $\tau$. For evaluation, the targeted adversarial attacks are variants of the PGD adversarial attack, i.e., PGD (Random) and PGD (Least Likely). PGD (Random) indicates that the target labels are chosen randomly. PGD (Least Likely) indicates that the target label is the

Table 9: Experimental results of the ablation study regarding the soft margin and the temperature parameter $\tau$. The baseline model is ResNet18 and the benchmark dataset is CIFAR-10.

| Methods | Natural (%) | Random | | Least Likely | |
|---|---|---|---|---|---|
| | | PGD$^{20}$ (%) | PGD$^{50}$ (%) | PGD$^{20}$ (%) | PGD$^{50}$ (%) |
| Pure Margin | 84.53 | 56.21 | 56.04 | 75.45 | 75.24 |
| Soft Margin ($\tau$=0.05) | 85.88 | 63.28 | 63.20 | 77.84 | 77.65 |
| Soft Margin ($\tau$=0.5) | 85.96 | 69.32 | 68.86 | 82.07 | 81.94 |
| Soft Margin ($\tau$=1) | 85.88 | 70.00 | 69.31 | 82.06 | 81.84 |
| Soft Margin ($\tau$=5) | **86.32** | **72.75** | **72.35** | **83.31** | **83.28** |
| Soft Margin ($\tau$=50) | 86.25 | 71.20 | 70.61 | 82.20 | 82.02 |
| Soft Margin ($\tau$=500) | **86.32** | 69.55 | 69.53 | 80.75 | 80.44 |

label corresponding to the smallest output logit. Both attack strategies for selecting target labels are designed following the convention [32]. As shown in Table 9, the adversarial accuracy under targeted adversarial attacks increases when the pure margin is changed to the soft margin. Consequently, the soft margin is shown to be effective to improve the adversarial accuracy under the targeted adversarial attack. Furthermore, when the temperature parameter $\tau$ is set to five, the adversarial accuracy becomes the highest, demonstrating the validity of setting $\tau$ to five as default.

Table 10: Experimental results of the ablation study in terms of different adversarial training frameworks. The baseline model is ResNet18 and the benchmark dataset is CIFAR-10.

| Methods | Natural (%) | Adversarial Attacks | | | Training Cost |
|---|---|---|---|---|---|
| | | FGSM (%) | PGD$^{20}$ (%) | PGD$^{50}$ (%) | (GPU Hours) |
| TRADES [56] | 81.16 | 60.27 | 52.45 | 52.39 | 6.44 |
| TRADES + VDAT | **81.43** | **61.15** | **55.55** | **55.50** | **3.27** |
| AWP [49] | 81.97 | 58.47 | 53.98 | 53.91 | 7.15 |
| AWP + VDAT | **83.97** | **62.32** | **56.03** | **55.93** | **3.76** |

In addition, we perform ablation studies to demonstrate the flexibility of the proposed VDAT method. In particular, we integrate VDAT into AWP [49] and TRADES [56] adversarial training frameworks. In this set of experiments, the benchmark dataset is CIFAR-10 and the deep neural network for training is ResNet18. The natural accuracy, adversarial accuracy, and training cost are reported in Table 10. As can be seen, VDAT can improve the natural accuracy and adversarial accuracy of both TRADES and AWP. Meanwhile, the training cost of both TRADES and AWP is decreased owing to the integration of VDAT. In summary, VDAT is shown to enjoy the flexibility across different adversarial training frameworks.

Please note that additional experimental results regarding the parameter study and ablation study can be found in **Appendix E**.

## 6 Conclusion

In this paper, we mainly focus on filtering the vulnerable data in the adversarial training process. This goal is achieved by the proposed VDAT method with two core components, i.e., the margin-based vulnerability calculation and vulnerability-aware data filtering. The experimental results demonstrate the effectiveness and scalability of VDAT. In addition, the validity of core components designed in VDAT is shown by the ablation studies. In the future, we will put the effort into theoretically analyzing the effectiveness of the proposed VDAT method.

## Acknowledgments

This work was supported by National Natural Science Foundation of China under Grant 62276175 and Innovative Research Group Program of Natural Science Foundation of Sichuan Province under Grant 2024NSFTD0035.

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

# Appendix

The appendix contains contents from following aspects, in order to provide more insights about the proposed VDAT method:

- **Appendix A**: This section presents additional details about the methodology of VDAT.

- **Appendix B**: This section contains additional details of experimental settings to facilitate the reproducibility.

- **Appendix C**: This section presents experimental results regarding robust NAS, containing the results of VDAT on CIFAR-100 and ImageNet-1K, the results in different search spaces, the results under black-box attacks, and the visualizations of the search process.

- **Appendix D**: This section contains additional visualizations of VDAT, i.e., the visualizations regarding the training process on ImageNet-1K and the normalized margin differences.

- **Appendix E**: This section presents additional results in terms of the parameter studies and ablation studies.

## A More Details about the Workflow of VDAT

To help readers better understand how VDAT is integrated into robust NAS methods, we present a framework of robust NAS methods along with VDAT. The workflow is presented in Algorithm 2.

---

**Algorithm 2** General Workflow of Robust NAS with VDAT

---

**Input:** Target dataset $\mathcal{X} = \{x_1, x_2, \ldots, x_n\}$, the pre-determined nodes and operations, the number of epochs $\mathcal{N}$ for search, and the interval $\mathcal{T}$
**Output:** The derived robust neural architecture

1: Initialize the supernet based on the pre-determined nodes and operations
2: Initialize the trainable weights $\omega$ and architecture parameters $\alpha$ of the supernet
3: Divide $\mathcal{X}$ into two halves, i.e., $\mathcal{X}^{\text{Train}}$ and $\mathcal{X}^{\text{Valid}}$
4: **for** $i = 0$ to $\mathcal{N}$ **do**
5:     **if** $i \mod \mathcal{T} == 0$ **then**
6:        Calculate $\mathcal{P}_{\omega,\alpha}(x_i)$ for each sample in $\mathcal{X}^{\text{Train}}$ and $\mathcal{X}^{\text{Valid}}$
7:        Perform data filtering and update corresponding subsets $\mathcal{X}_{\text{nat}}^{\text{Train}}$, $\mathcal{X}_{\text{adv}}^{\text{Train}}$, $\mathcal{X}_{\text{nat}}^{\text{Valid}}$, and $\mathcal{X}_{\text{adv}}^{\text{Valid}}$
8:     **end if**
9:     Calculate $\mathcal{L}_{\text{train}} = \mathcal{L}_{\text{nat}}^{\text{train}}(\mathcal{X}_{\text{nat}}^{\text{Train}}, \omega, \alpha) + \mathcal{L}_{\text{adv}}^{\text{train}}(\mathcal{X}_{\text{adv}}^{\text{Train}}, \omega, \alpha)$
10:     Keep the architecture parameter frozen, and update the trainable weights $\omega$ based on $\mathcal{L}_{\text{train}}$
11:     Keep the trainable weights $\omega$ frozen, update the architecture parameter $\alpha$ according to the update strategy of this robust NAS method based on $\mathcal{X}_{\text{nat}}^{\text{Valid}}$ and $\mathcal{X}_{\text{adv}}^{\text{Valid}}$
12: **end for**
13: Determine the searched architecture based on the architecture parameter $\alpha$ optimized
14: **Return** The derived robust neural architecture

---

As shown in Algorithm 2, given the target dataset $\mathcal{X}$, along with the pre-determined nodes, operations, the number $\mathcal{N}$ of search epoch, and the interval $\mathcal{T}$, the robust neural architecture is derived finally. There are four parts (lines 1-3, lines 5-8, lines 9-11, and line 13) in total in the search process. In the fist part, the supernet along with the trainable weights $\omega$ and the architecture parameters $\alpha$ are initialized (lines 1-2). Then, the target dataset $\mathcal{X}$ is equally divided into two parts, i.e., $\mathcal{X}^{\text{Train}}$ and $\mathcal{X}^{\text{Valid}}$ for updating $\omega$ and $\alpha$, respectively (line 3). In the second part, the data filtering is performed according to the interval $\mathcal{T}$, and $\mathcal{X}^{\text{Train}}$ and $\mathcal{X}^{\text{Valid}}$ are filtered respectively (lines 6-7). In the third part, the trainable weights is updated based on the training loss $\mathcal{L}_{\text{train}}$ calculated (lines 9-10). Then, the architecture parameters $\alpha$ is updated based on the update strategy of this robust NAS method (line 11). In the fourth part, the robust neural architecture is determined based on the architecture parameter $\alpha$ optimized (line 13).

# B More Details of Experimental Settings

In this section, we present additional details about the experimental settings. Specifically, the details are mainly from three aspects, i.e., the experimental settings for the adversarial training, robust NAS, and evaluations.

## B.1 Experimental Settings for Adversarial Training

We determine the experimental settings by following the conventions in the FAT community [28, 27]. For the adversarial training on CIFAR-10 and CIFAR-100, the total epoch is set to 110. Meanwhile, the batch size is set to 128, the learning rate is set to 0.1, the momentum is set to 0.9, and the weight decay is set to $10^{-4}$. The learning rate is decayed by the factor 0.1 at the 100-th and 105-th epochs, respectively. As for the adversarial training on ImageNet-1K, the total epoch is set to 60. Besides, the batch size is set to 512, while the momentum, the weight decay, and the learning rate are set to the same values as those on CIFAR-10 and CIFAR-100. The learning rate is decayed by the factor 0.1 at the 20-th and 40-th epochs, respectively. Please note that all the experiments are performed on the NVIDIA RTX 3090 GPU.

## B.2 Experimental Settings for Robust NAS

The total number of epoch for search is set to 50 for all four robust NAS methods. As for the other hyperparameters, we following the original settings of these robust NAS methods in their seminal papers [36, 24, 41, 38].

## B.3 Experimental Settings for Evaluations

The experimental setting for evaluations consists of two parts. The first part mainly focuses on the adversarial training of neural architectures derived by robust NAS methods. The second part mainly focuses on the settings of adversarial attacks for evaluations.

### B.3.1 Settings for Training Architectures Searched

The experimental settings for the adversarial training follow the convention in the previous study [22, 17, 19]. The PGD adversarial training is adopted to train the derived architectures [35] on CIFAR-10, CIFAR-100, and ImageNet-16-120 [7]. Besides, the FGSM-based FAT method [48] is adopted to train neural architectures on ImageNet-1K. The detailed experimental settings for adversarial training are presented in Table 11.

Table 11: The experimental settings for the adversarial training on CIFAR-10, CIFAR-100, ImageNet-16-120, and ImageNet-1K.

| Items | CIFAR | ImageNet-16-120 | ImageNet-1K |
|---|---|---|---|
| Optimizer | SGD | SGD | SGD |
| Momentum | 0.9 | 0.9 | 0.9 |
| Epochs | 200 | 200 | 40 |
| Learning Rate | 0.1 | 0.1 | 0.1 |
| Learning Rate Decay | (100, 150) | (100, 150) | (20, 30) |
| Weight Decay | $10^{-4}$ | $10^{-4}$ | $10^{-4}$ |
| PGD Step | 7 | 7 | / |
| FGSM $\epsilon$ | / | / | 4/255 |
| PGD $\epsilon$ | 8/255 | 8/255 | / |
| PGD Step Size | 0.01 | 0.01 | / |

### B.3.2 Settings for Adversarial Attacks

Furthermore, the experimental settings for the adversarial attacks also follow the convention [41, 38]. The details for the adopted FGSM [21], PGD [35], C&W [2], and AutoAttack (AA) [10] adversarial attacks are shown in Table 12.

Table 12: The experimental settings for the adversarial attacks FGSM, PGD, C&W, and AA.

| Items | FGSM | PGD | C&W | AA |
|---|---|---|---|---|
| Total Perturbation Scale $\epsilon$ | 8/255 | 8/255 | / | 8/255 |
| Step Size | / | 2/255 | / | / |
| Steps | / | 20, 50, 100 | 100 | / |
| $c$ | / | / | 0.5 | / |

## C  Additional Experimental Results for Robust NAS

To further demonstrate the effectiveness of VDAT in the scenario of robust NAS, we conduct experiments in terms of evaluations under white-box attacks, black-box attacks, different search spaces, and visualizations of the search processes. These experimental results are presented in order in the following sections.

### C.1  Experiments under White-box Attacks

Beyond the experimental results under white-box attacks on CIFAR-10 (Table 4), we also conduct experiments on CIFAR-100 and ImageNet-1K, to comprehensively evaluate the effectiveness of VDAT on both medium-scale and large-scale datasets. The robust neural architectures are searched and then adversarially trained for evaluations. The experimental results on CIFAR-100 and ImageNet-1K are presented in Tables 13 and 14, respectively.

Table 13: Experimental results of VDAT on CIFAR-100 under white-box attacks. The adversarial training methods are integrated into the corresponding robust NAS methods.

| Models | Params | Natural (%) | Adversarial Attacks | | | | | Search Cost |
|---|---|---|---|---|---|---|---|---|
| | | | FGSM (%) | PGD$^{20}$ (%) | PGD$^{100}$ (%) | C&W (%) | AA (%) | (GPU Days) |
| DSRNA [24] | 2.0M | 57.44 | 35.03 | 28.11 | 27.97 | 21.52 | 25.20 | 0.80 |
| + FGSM-SDI [28] | 2.0M | 60.18 | 28.67 | 22.50 | 22.24 | _24.63_ | 21.07 | 0.53 |
| + FGSM-PGK [27] | 2.8M | 60.35 | 31.26 | 26.54 | 26.38 | **27.99** | 24.25 | _0.43_ |
| **+ VDAT**$_{PGD}$ | 3.2M | **63.82** | **38.79** | _31.32_ | _30.89_ | 24.22 | _28.48_ | 0.65 |
| **+ VDAT**$_{FGSM}$ | 3.5M | _61.02_ | _37.41_ | **33.31** | **33.07** | 22.12 | **31.42** | **0.34** |
| AdvRush [36] | 4.2M | 58.73 | 39.51 | 30.15 | 29.67 | 20.08 | 26.46 | 1.22 |
| + FGSM-SDI [28] | 3.3M | 59.09 | 34.03 | 29.18 | 28.80 | **34.98** | 25.96 | 0.74 |
| + FGSM-PGK [27] | 3.7M | 61.01 | 31.41 | 25.73 | 25.34 | 24.94 | 24.44 | _0.59_ |
| **+ VDAT**$_{PGD}$ | 3.5M | _61.51_ | _40.98_ | **32.44** | **31.47** | _28.60_ | **30.29** | 0.76 |
| **+ VDAT**$_{FGSM}$ | 3.7M | **61.91** | **41.40** | _31.04_ | _30.81_ | 20.87 | _29.13_ | **0.47** |
| RNAS [41] | 3.5M | _60.24_ | 40.52 | 31.11 | 31.06 | 20.84 | 27.37 | 0.71 |
| + FGSM-SDI [28] | 4.4M | 59.60 | 30.83 | 24.99 | 24.63 | 24.80 | 23.07 | 0.42 |
| + FGSM-PGK [27] | 3.4M | 59.53 | 32.74 | 26.67 | 26.37 | _28.63_ | 25.08 | _0.34_ |
| **+ VDAT**$_{PGD}$ | 4.1M | 58.73 | _41.45_ | _33.14_ | _32.77_ | **29.01** | _30.20_ | 0.68 |
| **+ VDAT**$_{FGSM}$ | 3.8M | **61.07** | **43.21** | **34.60** | **34.37** | 21.62 | **30.82** | **0.27** |
| ARNAS [38] | 4.5M | 58.18 | 32.60 | 29.54 | 29.30 | 19.51 | 25.76 | 2.95 |
| + FGSM-SDI [28] | 5.0M | 62.00 | 32.54 | 26.48 | 26.07 | _24.66_ | 24.38 | 1.81 |
| + FGSM-PGK [27] | 3.8M | _63.40_ | 32.45 | 26.30 | 26.06 | 20.54 | 25.35 | _1.44_ |
| **+ VDAT**$_{PGD}$ | 5.7M | 62.78 | _33.54_ | **31.44** | **30.96** | **26.95** | **29.62** | 2.22 |
| **+ VDAT**$_{FGSM}$ | 5.0M | **63.43** | **35.32** | _29.84_ | _29.59_ | 21.82 | _28.46_ | **1.15** |

As can be seen from Table 13, VDAT$_{FGSM}$ consistently achieves better natural accuracy compared with state-of-the-art FAT methods and the baselines. Meanwhile, VDAT$_{PGD}$ can achieve the state-of-the-art natural accuracy in the cases of DSRNA and AdvRush. Moreover, after combining VDAT with robust NAS methods selected, the adversarial accuracy reaches the highest in most cases. Furthermore, VDAT$_{FGSM}$ also consistently achieves the lowest search cost compared with the cases of state-of-the-art FAT methods and vanilla robust NAS methods. In addition, the robust NAS methods integrated with VDAT$_{PGD}$ also demonstrates better efficiency than the original ones.

As shown in Table 14, both natural accuracy and adversarial accuracy are improved after introducing VDAT$_{FGSM}$. These results demonstrate the effectiveness of VDAT in terms of the large-scale dataset in the scenario of robust NAS.

Table 14: The experimental results of VDAT on ImageNet-1K under white-box attacks. VDAT is integrated into robust NAS methods.

| Models | Natural (%) | FGSM (%) | PGD$^{20}$ (%) | AA (%) |
|---|---|---|---|---|
| DSRNA [24] | 43.32 | 13.04 | 7.88 | 6.49 |
| + VDAT$_{FGSM}$ | **53.72** | **20.16** | **12.20** | **10.55** |
| AdvRush [36] | 51.54 | 18.42 | 10.74 | 9.23 |
| + VDAT$_{FGSM}$ | **54.88** | **21.12** | **12.67** | **11.15** |
| RNAS [41] | 51.18 | 17.62 | 9.87 | 8.61 |
| + VDAT$_{FGSM}$ | **53.26** | **20.15** | **11.85** | **10.29** |
| ARNAS [38] | 54.69 | 20.60 | 11.27 | 11.13 |
| + VDAT$_{FGSM}$ | **57.05** | **22.68** | **13.14** | **12.90** |

## C.2 Experiments under Black-box Attacks

To evaluate the architectures derived by robust NAS methods more comprehensively, we conduct experiments under black-box adversarial attacks beyond white-box adversarial attacks. In this set of experiments, we choose the commonly used transfer-based black-box attack [6] to further validate the adversarial robustness of the searched architectures. Specifically, the adversarial examples are generated based on the PGD$^{20}$ adversarial attack, and then they are transferred to attack other models. This practice follows the convention [36, 38]. The experimental results under the black-box attack are presented in Table 15. As can be seen, when VDAT is integrated into robust NAS methods, the adversarial accuracy under the black-box attack still improves, demonstrating its effectiveness under the black-box attack.

Table 15: The experimental results under the transfer-based black-box attack on CIFAR-10 and CIFAR-100. The highlighted data means that the source model and the target model are the same, thus the experimental results are the same as those under the white-box attack.

| Datasets | Target / Source | AdvRush | RNAS | ARNAS | AdvRush + VDAT$_{FGSM}$ | RNAS + VDAT$_{FGSM}$ | ARNAS + VDAT$_{FGSM}$ |
|---|---|---|---|---|---|---|---|
| CIFAR-10 | AdvRush | 53.07 | 68.08 | 71.46 | 74.03 | 73.76 | 75.95 |
| | RNAS | 68.61 | 53.48 | 66.99 | 72.87 | 70.92 | 74.89 |
| | ARNAS | 66.90 | 62.22 | 55.87 | 71.51 | 70.05 | 72.86 |
| | AdvRush + VDAT$_{FGSM}$ | 67.43 | 65.66 | 68.88 | 55.32 | 70.41 | 73.18 |
| | RNAS + VDAT$_{FGSM}$ | 68.40 | 64.65 | 68.52 | 71.69 | 55.43 | 73.42 |
| | ARNAS + VDAT$_{FGSM}$ | 66.90 | 62.22 | 55.86 | 71.55 | 70.00 | 57.15 |
| CIFAR-100 | AdvRush | 30.15 | 36.55 | 39.27 | 43.37 | 43.72 | 46.24 |
| | RNAS | 41.55 | 31.11 | 40.22 | 45.61 | 44.83 | 47.47 |
| | ARNAS | 41.14 | 36.89 | 29.54 | 45.14 | 44.38 | 46.91 |
| | AdvRush + VDAT$_{FGSM}$ | 41.03 | 38.06 | 40.89 | 31.04 | 42.77 | 46.23 |
| | RNAS + VDAT$_{FGSM}$ | 46.50 | 42.16 | 45.16 | 47.86 | 34.60 | 49.18 |
| | ARNAS + VDAT$_{FGSM}$ | 41.17 | 36.91 | 29.61 | 45.04 | 44.41 | 29.84 |

## C.3 Experiments in Different Search Spaces

The above experimental results are all performed in a classical search space in the NAS community, i.e., the DARTS search space [34]. To further demonstrate the scalability of VDAT in the scenario of robust NAS, we perform experiments on both NAS-Bench-101 [52] and NAS-Bench-201 [15] search spaces. As can be seen in Table 16, VDAT can still bring improvements regarding both natural accuracy and adversarial accuracy in most cases. Please note that the improvement on NAS-Bench-201 is much more significant than that on NAS-Bench-101. This is because the search space of NAS-Bench-201 is much smaller than NAS-Bench-101, so that the architecture parameter can overfit the search data more easily. This will lead to the enrichment of the "skip connection" operation, causing the performance collapse [46]. Fortunately, VDAT can mitigate such overfitting because the data for training and search is changed iteratively during the search process, thus VDAT brings significant improvements.

Table 16: Experimental results of VDAT with robust NAS methods on NAS-Bench-101 and NAS-Bench-201 benchmarks. The datasets are CIFAR-10, CIFAR-100, and ImageNet-16-120.

| | **NAS-Bench-101** | | | | | | | | | | | |
| --- | --- | --- | --- | --- | --- | --- | --- | --- | --- | --- | --- | --- |
| | **CIFAR-10** | | | | **CIFAR-100** | | | | **ImageNet-16-120** | | | |
| **Models** | Nat. | FGSM | PGD$^{20}$ | AA | Nat. | FGSM | PGD$^{20}$ | AA | Nat. | FGSM | PGD$^{20}$ | AA |
| AdvRush [36] | 82.93 | 51.19 | 40.14 | 41.41 | 55.45 | 25.33 | 20.27 | 20.36 | **17.55** | 5.90 | 4.77 | 3.57 |
| + VDAT$_{FGSM}$ | **85.03** | **56.24** | **47.07** | **46.28** | **58.23** | **30.17** | **25.86** | **24.53** | 17.28 | **7.25** | **6.52** | **3.60** |
| RNAS [41] | 83.28 | 50.93 | 39.41 | 41.14 | 55.19 | 26.28 | 21.64 | 21.30 | 17.52 | 5.75 | 4.45 | **3.52** |
| + VDAT$_{FGSM}$ | **85.05** | **56.61** | **47.50** | **46.82** | **59.33** | **28.75** | **23.81** | **22.81** | **17.62** | **7.15** | **6.43** | 3.48 |
| **ARNAS** [38] | 82.46 | 54.91 | **48.80** | 46.57 | 54.22 | 29.79 | 26.09 | 23.78 | 17.52 | 6.73 | 5.78 | 4.05 |
| + VDAT$_{FGSM}$ | **84.08** | **55.93** | 47.89 | **46.72** | **56.37** | **29.99** | **26.61** | **24.10** | **17.77** | **7.22** | **6.23** | **4.35** |
| | **NAS-Bench-201** | | | | | | | | | | | |
| | **CIFAR-10** | | | | **CIFAR-100** | | | | **ImageNet-16-120** | | | |
| **Models** | Nat. | FGSM | PGD$^{20}$ | AA | Nat. | FGSM | PGD$^{20}$ | AA | Nat. | FGSM | PGD$^{20}$ | AA |
| AdvRush [36] | 59.31 | 35.58 | 33.26 | 30.94 | 30.17 | 15.52 | 14.71 | 12.65 | 9.58 | 5.30 | 5.13 | **3.47** |
| + VDAT$_{FGSM}$ | **85.06** | **59.43** | **51.75** | **50.10** | **58.41** | **30.67** | **26.20** | **24.72** | **16.42** | **8.78** | **8.50** | 2.75 |
| RNAS [41] | 73.29 | 45.81 | 41.32 | 40.13 | 40.13 | 20.66 | 19.20 | 16.95 | 13.20 | 6.55 | 6.02 | **4.17** |
| + VDAT$_{FGSM}$ | **82.68** | **56.90** | **49.63** | **47.81** | **55.33** | **28.68** | **24.82** | **23.12** | **17.13** | **8.37** | **7.68** | 4.12 |
| **ARNAS** [38] | 79.91 | 54.05 | 48.39 | 46.14 | 50.47 | 26.32 | 23.49 | 21.25 | 16.48 | 6.88 | 6.00 | 4.18 |
| + VDAT$_{FGSM}$ | **84.84** | **58.56** | **51.19** | **49.41** | **56.95** | **29.90** | **25.74** | **23.75** | **17.48** | **7.28** | **6.57** | **4.75** |

### C.4   Visualizations of the Search Process

Because the data filtering method in VDAT changes the search data iteratively in the search process, we also explore the convergence of the search to reveal the reliability of VDAT in the scenario of robust NAS. Specifically, there are four indicators reported to show the convergence, i.e., the training and validation accuracy for the data in $\mathcal{X}_{nat}^{Train}$, $\mathcal{X}_{nat}^{Valid}$ and the data in $\mathcal{X}_{adv}^{Train}$, $\mathcal{X}_{adv}^{Valid}$. The experimental results are shown in Figure 6. As can be seen, all four indicators converge effectively in the search process, demonstrating that VDAT will not affect the convergence of the robust NAS methods. Please note that the indicators drop at the 40-th epoch in Figure 6a. This is caused by the regularization in AdvRush and in line with the phenomenon in its original paper.

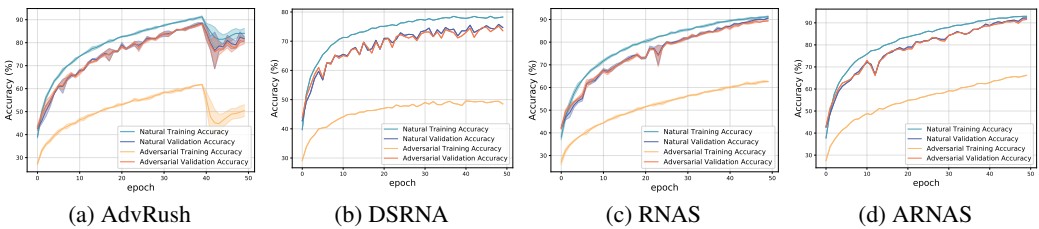

|  (a) AdvRush  |  (b) DSRNA  |  (c) RNAS  |  (d) ARNAS  |

Figure 6: Visualization for the search process of robust NAS methods after introducing VDAT.

## D   Additional Visualizations for Adversarial Training

In this section, we present additional visualizations regarding the adversarial training. The visualizations come from two aspects which are same as those in the main text, i.e., the training process and the vulnerability of data samples. The details are presented in the following sections.

### D.1   Visualizations of the Training Process on ImageNet-1K

Beyond the visualizations of the training process on CIFAR-10 and CIFAR-100, we also visualize the training process of VDAT$_{FGSM}$ on ImageNet-1K in Figure 7. The natural accuracy and the adversarial accuracy on the test set are visualized. The adversarial accuracy is evaluated by the PGD$^{10}$ adversarial attack. As can be seen from Figure 7, even though the data for natural training and adversarial training changes during the whole training process, both natural accuracy and adversarial accuracy converge

well in the training process. More importantly, the catastrophic overfitting phenomenon also does not occur in the training process on ImageNet-1K. These observations indicate that VDAT enjoys promising convergence on the large-scale dataset, i.e., ImageNet-1K. Meanwhile, VDAT also does not suffer from the catastrophic overfitting on the large-scale dataset, demonstrating the decent reliability of VDAT.

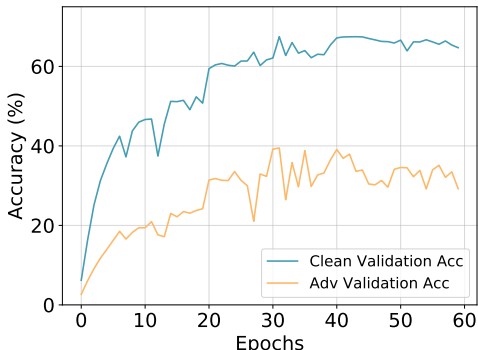

Figure 7: Visualization of the training processes of VDAT$_{\text{FGSM}}$ on ImageNet-1K.

### D.2 Visualizations of the Vulnerability of Data Samples

In this part, we present additional visualizations of the vulnerability of data samples. Specifically, we visualize the normalized margin difference (larger margin difference indicate lower vulnerability) and the corresponding number of data samples on CIFAR-10 and CIFAR-100. The baseline model correlated to these results is WideResNet34-10 and the training method is VDAT$_{\text{FGSM}}$. The visualizations are presented in Figure 8. As can be seen, the distribution shifts to right after the training process of VDAT$_{\text{FGSM}}$ on both CIFAR-10 and CIFAR-100. This phenomenon indicates VDAT$_{\text{FGSM}}$ can indeed lower the vulnerability of data samples when the baseline model is WideResNet34-10.

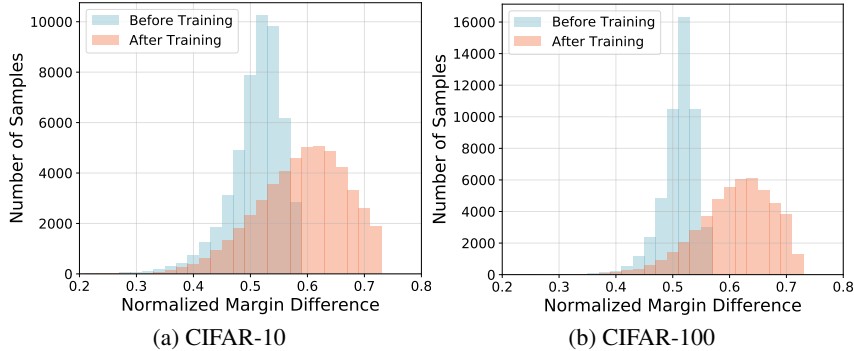

(a) CIFAR-10          (b) CIFAR-100

Figure 8: Visualizations of the normalized margin difference of data before and after training by VDAT$_{\text{FGSM}}$. The baseline model is WideResNet34-10.

Besides, we also conduct the same kind of visualizations for the adversarial training method VDAT$_{\text{PGD}}$ in Figure 9. It is obvious that the visualizations in Figure 9 are similar to those in Figures 5 and 8. Therefore, it is illustrated that VDAT$_{\text{PGD}}$ can also lower the vulnerability of data samples after training, thus the effectiveness of VDAT is further demonstrated.

## E   Additional Parameter Studies and Ablation Studies

In this section, we present additional parameter studies and ablation studies to further demonstrate the effectiveness of the designed components in VDAT. These results are divided into four parts, i.e., the parameter study and ablation study in the scenario of robust NAS, the ablation study regarding

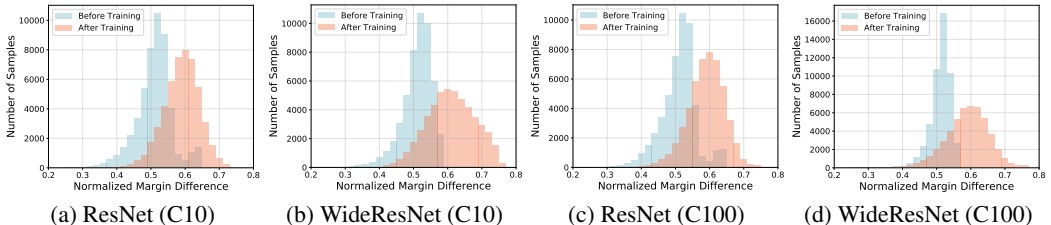

| (a) ResNet (C10) | (b) WideResNet (C10) | (c) ResNet (C100) | (d) WideResNet (C100) |

Figure 9: Visualizations of the normalized margin difference of data before and after training by VDAT$_{\text{PGD}}$.

the adversarial loss $\mathcal{L}_{\text{adv}}$ and the natural loss $\mathcal{L}_{\text{nat}}$, and the ablation study in terms of the vulnerability assessment.

### E.1 Parameter Study in the Scenario of Robust NAS

In this part, we conduct the hyperparameter study for the hyperparameter $\tau$ in the scenario of robust NAS. The experimental results are shown in Table 17. As can be seen, no matter which value $\tau$ is set to, the natural accuracy and adversarial accuracy are constantly improved compared with the baseline robust NAS method DSRNA. When $\tau$ is set to five, both natural accuracy and adversarial accuracy achieves the best. Therefore, the validity of our default setting of $\tau$ is demonstrated.

Table 17: The results of the parameter study of hyperparameter $\tau$ under robust NAS settings. The benchmark dataset is CIFAR-10, and the backbone robust NAS method is DSRNA.

| $\tau$ | Params | Natural (%) | FGSM (%) | PGD$^{20}$ (%) |
|---|---|---|---|---|
| DSRNA | 2.0M | 80.93 | 54.49 | 49.11 |
| 0.05 ($\times 0.01$) | 3.6M | 87.69 | 60.33 | 51.67 |
| 0.5 ($\times 0.1$) | 2.8M | 87.49 | 60.73 | 51.71 |
| **5 (Default)** | 3.4M | **91.10** | **84.23** | **64.01** |
| 50 ($\times 10$) | 3.5M | 87.51 | 61.42 | 52.62 |
| 500 ($\times 100$) | 3.4M | 87.94 | 60.59 | 50.53 |

### E.2 Ablation Study in the Scenario of Robust NAS

To validate the effectiveness of data filtering under robust NAS settings, we validate three situations in this set of experiments, the vanilla robust NAS (DSRNA), robust NAS with random filtering, and the default vulnerability-aware data filtering in VDAT. As shown in Table 18, the random filtering can bring improvements regarding the natural accuracy and adversarial robustness of the derived architecture, but they are still inferior to the vulnerability-aware data filtering. These observations are in line with those in Table 7. In summary, the effectiveness of the vulnerability-aware data filtering is demonstrated.

Table 18: The ablation study under the robust NAS settings. The backbone robust NAS method is DSRNA, and the benchmark dataset is CIFAR-10.

| | | | Adversarial Attacks | | | |
|---|---|---|---|---|---|---|
| **Methods** | **Params** | **Natural (%)** | **FGSM (%)** | **PGD$^{20}$ (%)** | **PGD$^{100}$ (%)** | **AA (%)** |
| DSRNA (Without Filtering) | 2.0M | 80.93 | 54.49 | 49.11 | 48.89 | 44.87 |
| + VDAT (Random Filtering) | 3.0M | 87.52 | 60.66 | 51.68 | 50.90 | 47.45 |
| **+ VDAT** | 3.4M | **91.10** | **84.23** | **64.01** | **54.64** | **50.26** |

### E.3 Ablation Study of $\mathcal{L}_{\text{adv}}$ and $\mathcal{L}_{\text{nat}}$

In order to validate the necessity of optimizing both $\mathcal{L}_{\text{adv}}$ and $\mathcal{L}_{\text{nat}}$ instead of single $\mathcal{L}_{\text{adv}}$ in Equation (8), we conduct the ablation study regarding $\mathcal{L}_{\text{adv}}$ and $\mathcal{L}_{\text{nat}}$. In the situation of optimizing $\mathcal{L}_{\text{adv}}$

Table 19: Ablation study regarding the natural loss $\mathcal{L}_{adv}$ and the adversarial loss $\mathcal{L}_{nat}$ on CIFAR-10. The baseline model is WideResNet34-10, and the adversarial perturbation for adversarial training is generated by PGD adversarial attack.

| $\mathcal{L}_{adv}$ | $\mathcal{L}_{nat}$ | Natural (%) | PGD$^{50}$ (%) | Cost |
|---|---|---|---|---|
| ✓ | ✗ | 79.99 | 55.00 | **8.67** |
| ✓ | ✓ | **88.10** | **60.26** | 9.77 |

individually, we directly remove the data samples filtered for natural training, and only train the model with samples filtered for adversarial training. The experimental results are shown in Table 19. As can be seen, when optimizing $\mathcal{L}_{adv}$ individually, both natural accuracy and adversarial accuracy drop. This is mainly because the useful features which are contained by data samples filtered for natural training are removed, thus the overall performance is negatively affected. Moreover, the training cost decreases slightly when removing $\mathcal{L}_{nat}$, this is because the natural training process is removed. However, because the natural training does not cost much computational budget, we still choose to optimize $\mathcal{L}_{adv}$ and $\mathcal{L}_{nat}$ at the same time to ensure the promising natural accuracy and adversarial accuracy.

### E.4 Ablation Study of the Vulnerability Assessment

This set of experiments consists of two parts. First, we evaluate different types of margins for vulnerability assessment under the non-targeted adversarial attacks. Second, we compare different adversarial attacks for the vulnerability assessment.

In the first part, we evaluate the pure margin (shown in Equation (2)) and the soft margin under the non-targeted adversarial attacks. The benchmark datasets are CIFAR-10 and CIFAR-100, and the baseline model for adversarial training is ResNet18. As shown in Table 20, both natural accuracy and adversarial accuracy decrease when changing the soft margin to the pure margin. Consequently, the proposed margin-based vulnerability calculation method is shown to be effective under non-targeted adversarial attacks.

Table 20: Ablation study regarding different types of margins for vulnerability assessment. The adversarial attacks are non-targeted attacks.

| Datasets | Margins | Natural (%) | FGSM (%) | PGD$^{20}$ (%) | PGD$^{50}$ (%) |
|---|---|---|---|---|---|
| CIFAR-10 | Pure Margin | 84.53 | 58.61 | 52.30 | 52.16 |
| CIFAR-10 | Soft Margin | **86.32** | **62.94** | **56.30** | **56.23** |
| CIFAR-100 | Pure Margin | 58.36 | 37.74 | 27.58 | 27.46 |
| CIFAR-100 | Soft Margin | **59.43** | **41.17** | **35.69** | **35.61** |

Table 21: Ablation study regarding different adversarial attacks for vulnerability assessment. The loss for adversarial training is the PGD loss.

| Perturbation | Natural (%) | PGD$^{50}$ (%) | Cost (GPU Hours) |
|---|---|---|---|
| PGD$^{10}$ | 82.15 | 58.35 | 3.04 |
| PGD$^{20}$ | 81.99 | **58.80** | 3.75 |
| FGSM | **82.96** | 56.07 | **1.96** |

In the second part, we evaluate PGD$^{10}$, PGD$^{20}$, and FGSM adversarial attacks for vulnerability assessment. Different from the FGSM training loss adopted in Table 8, this set of experiments adopts PGD training loss for a comprehensive evaluation. As shown in Table 21, using stronger attacks (i.e., PGD$^{10}$ and PGD$^{20}$) for vulnerability assessment can increase the adversarial accuracy. However, this will also decrease the natural accuracy and training efficiency. Considering the balance of natural accuracy, adversarial accuracy, and training cost, we choose FGSM as the default attack for vulnerability assessment.

