# OpenReview forum: "Vulnerable Data-Aware Adversarial Training"
_NeurIPS.cc/2025/Conference — NeurIPS 2025 poster_

### Official Review · Reviewer_25Wa · 2025-06-23

**Clarity:** 2
**Significance:** 3
**Originality:** 3
**Rating:** 5
**Confidence:** 4

**Summary:**

This paper addresses the efficiency of fast adversarial training (FAT) by proposing vulnerable data-aware adversarial training (VDAT), which employs sample-wise data filtering rather than the batch-wise filtering used in existing FAT methods. Building on prior research showing that certain samples contribute minimally to adversarial robustness, the authors propose prioritizing vulnerable samples during training to achieve faster convergence. The filtering mechanism relies on a margin-based vulnerability metric defined through soft margins considering all classes, which quantifies the model's vulnerability to specific samples.

**Questions:**

I find the proposed method and experimental results satisfactory and believe they contribute meaningfully to the field. These contributions would be valuable even without theoretical guarantees. However, as noted in the weaknesses section, I am concerned that the paper contains clear errors or overclaiming in its theoretical analysis, which could mislead readers. If the authors provide a clear commitment to addressing these theoretical issues, I would be willing to raise my rating to Accept.

**Ethical Concerns:**

["NO or VERY MINOR ethics concerns only"]

**Final Justification:**

As my concerns have been addressed, and as stated in my earlier review, I am increasing my score from 3 to 5.

**Limitations:**

Yes, but the authors addressed only a few in the checklist.

**Quality:**

3

**Strengths And Weaknesses:**

**Strengths**

The paper presents a simple and intuitive concept of sample-wise data filtering using margins, which offers high compatibility with various adversarial example generation methods and demonstrates good generalizability. The proposed method shows superior performance compared to existing approaches in terms of both speed and accuracy. The experimental setup is generally comprehensive and well-designed.

**Weaknesses**

**1. Effectiveness of soft margin and temperature parameter $\tau$**

While the paper introduces the soft margin and temperature parameter $\tau$ to enhance robustness against targeted attacks, the ablation study in Table 5 does not clearly demonstrate whether these components actually improve effectiveness against targeted attacks as intended.

**2. Theoretical analysis contains errors or lacks clarity**

I believe the theoretical analysis section requires substantial revision or removal. The analysis suffers from insufficient assumptions and unclear explanations in the proofs, making it difficult to assess the validity of the theoretical claims.

Lemma 4.1 lacks necessary assumptions. While $f_θ$ is described as "a deep neural network," the specific definition remains unclear. If any classifier is acceptable, it should be written as a general function as shown in line 114. If it must be a neural network, the architecture should be explicitly specified. Although this is borrowed from [48], as required by NeurIPS Checklist 3, it should be formally provided within this paper. Additional details such as the parameter update method for $\theta$ (gradient descent? flow? [48] may use gradient flow) and the domain and conditions for $x$ and $y$ should also be clarified.

Upon reviewing the proof in [48], I found that their proof relies on Assumption 5.1, which you must also reference in your paper. Unfortunately, this assumption is not explicitly stated in the arXiv version of [48], making it impossible for me to infer its content. At minimum, this appears to involve positive eigenvalues of the Gram matrix, which is neither a simple nor obvious assumption and likely requires strong constraints such as infinite width for neural network architectures. Data assumptions such as spherical distributions may also be necessary. Please refer to neural tangent kernel literature.

Proposition 4.2 also contains vague statements such as "Training the target model based on datasets $X_{nat}$ and $X_{adv}$." In an extreme case, even "training" with a loss function $\ell(x, y) = 0$ would be included under this description, yet your claim would not hold in such cases.

Additionally, the proof in Appendix B relies heavily on verbal explanations rather than mathematical formulations, making it difficult to follow. I would appreciate specific references to the relevant sections in [48] where equation (11) is mentioned, as well as mathematical formulations for the operations described in lines 508-512. The mathematical derivations leading to the conclusion should be more explicitly stated.

In summary, to support Lemma 4.1 and Proposition 4.2, you should provide detailed definitions of the network architecture and parameter update methods, along with accurate references to all assumptions used, as required by NeurIPS Checklist 3. These clarifications may necessitate substantial rewriting of the proofs.

Given recent theoretical research in adversarial training, proving the theoretical effectiveness of this loss function for deep neural networks appears extremely challenging and would likely require dozens of pages of rigorous proof. Proofs for linear classifiers or at most two-layer networks would be more realistic in scope.

Based on these concerns, I recommend either:

- Substantial revision of the theorem statements, claims, and proofs, along with review of potentially overclaimed statements (e.g., lines 11, 12, 66).
- Removal of the theoretical claims entirely.

**Minor Comments**

- Line 115: Missing braces {}.
- The notation V_{x_i} and V_θ(x_i) are used inconsistently throughout the text (e.g., around line 131) and should be unified.
- In Algorithm 1, the model parameter θ may be missing from the input and output specifications.

---

> ### Author Rebuttal · Authors · 2025-07-30
>
> Thank you greatly for dedicating your time and expertise to reviewing our paper. Your comments and suggestions are truly helpful to enhance our paper and have pointed out the direction of our future work. Please see our response below.
>
> > **W1: Effectiveness of soft margin and temperature parameter $\tau$ against targeted attacks.**
>
> Thank you for pointing this out, and we have conducted experiments to show the effectiveness of soft margin and temperature parameter $\tau$ against targeted attacks. Specifically, we train ResNet18 on CIFAR-10 with the pure margin (Equation (2)) and the soft margin with different values of $\tau$. For evaluation, the targeted adversarial attacks are variants of the PGD attack, i.e., PGD_Random and PGD_Least_Likely. PGD_Random indicates that the target labels are chosen randomly. PGD_Least_Likely indicates that the target label is the label corresponding to the smallest output logit. Both strategies for selecting target labels are designed following the convention [1]. As shown in the table below, the adversarial accuracy under targeted attacks increases when the pure margin is changed to the soft margin. Consequently, the soft margin is shown to be effective to improve the adversarial accuracy under the targeted adversarial attack. Moreover, when the temperature parameter $\tau$ is set to five, the adversarial accuracy becomes the highest, demonstrating the validity of setting $\tau$ to five as default.
>
> |Method|Natural (%)|PGD20_Random (%)|PGD50_Random (%)|PGD20_Least_Likely (%)|PGD50_Least_Likely (%)|
> |-|-|-|-|-|-|
> |Pure Margin|84.53|56.21|56.04|75.45|75.24|
> |Soft Margin ($\tau$=0.05)|85.88|63.28|63.20|77.84|77.65|
> |Soft Margin ($\tau$=0.5)|85.96|69.32|68.86|82.07|81.94|
> |Soft Margin ($\tau$=1)|85.88|70.00|69.31|82.06|81.84|
> |Soft Margin ($\tau$=5)|**86.32**|**72.75**|**72.35**|**83.31**|**83.28**|
> |Soft Margin ($\tau$=50)|86.25|71.20|70.61|82.20|82.02|
> |Soft Margin ($\tau$=500)|**86.32**|69.55|69.53|80.75|80.44|
>
> > **W2: Theoretical analysis.**
>
> We sincerely appreciate your meticulous review. Following your suggestion, we promise to remove the theoretical claims entirely in the next version of this work. Specifically, we will remove or change the following contents:
>
> + Remove the descriptions about the theoretical analysis in Abstract (lines 11-12).
>
> + Remove the claim about the theoretical analysis in Introduction (line 66).
>
> + Remove contents of the theoretical analysis in Section 4.3 (lines 201-213).
>
> + Remove the descriptions of the proofs in Appendix B (lines 468-469).
>
> + Remove the whole contents about theoretical analysis and proofs in Appendix B (lines 493-515).
>
> + Change the answer in line 709 of NeurIPS Paper Checklist 3 to [NA], and the justification in line 710 to "The paper does not include theoretical results".
>
> Furthermore, we are actively working towards providing a rigorous proof regarding the effectiveness of the proposed VDAT method. Specifically, we will provide the whole set of assumptions in terms of the constraints for the network architecture, the parameter update method, the domain and conditions of the input data and labels, and the positive eigenvalues of the Gram matrix. Besides, the corresponding definitions about the network architecture and update methods, along with formulated proofs will be also provided. Thanks again for your valuable suggestions, and we hope to present a rigorous theoretical analysis in terms of the linear classifier as a complete but separate study real soon.
>
> > **W3: Line 115: Missing braces {}.**
>
> We will change "1,2,...,k" in the subscript to "{1,2,...,k}" in line 115.
>
> > **W4: The notation V_{x_i} and V_θ(x_i) are used inconsistently throughout the text (e.g., around line 131) and should be unified.**
>
> We will change the symbol $V_{x_i}$ in line 131 to $V_{\theta} (x_i)$ for uniformity. Moreover, the symbols $V_{x_i}$ and $P_{x_i}$ in Figure 3 will be also changed to $V_{\theta} (x_i)$ and $P_{\theta} (x_i)$, respectively.
>
> > **W5: In Algorithm 1, the model parameter θ may be missing from the input and output specifications.**
>
> We will change "the target model" in Input to "the target model with parameter $\theta$", and "The trained model" in Output to "The trained model with parameter $\theta*$". Besides, the words "The trained model with $\theta$" in line 9 of Algorithm 1 will be changed to "The trained model with $\theta*$".
>
> **Reference**
>
> [1] Adversarial machine learning at scale, ICLR 2017.

---

> > ### Comment · Reviewer_25Wa · 2025-08-04
> >
> > Thank you for the author's response. As my concerns have been addressed, and as stated in my earlier review, I am increasing my score from 3 to 5.

---

### Official Review · Reviewer_sLg7 · 2025-06-25

**Clarity:** 3
**Significance:** 3
**Originality:** 3
**Rating:** 4
**Confidence:** 2

**Summary:**

The paper introduces Vulnerable Data-aware Adversarial Training (VDAT), a novel approach to fast adversarial training (FAT) by focusing on sample-wise data filtering rather than the traditional batch-wise methods, which can improve natural accuracy and adversarial accuracy while reducing adversarial training cost.

**Questions:**

1. Could you please explain the significant accuracy improvement observed around epoch 100 in Figure 4?
2. Could you please explain why VDAT_FGSM generally performs better than VDAT_SGD on both natural accuracy and adversarial accuracy? This seems a bit counterintuitive to me, since SGD provides stronger adversarial examples.
3. The vulnerability calculation relies on an initial perturbation generated by a single-step FGSM attack for efficiency. How sensitive are the results to this choice? Specifically, if a stronger (e.g., PGD) but more costly attack was used to assess vulnerability, would it identify a more correct set of vulnerable samples and lead to a more robust final model, even if the training itself still uses FGSM or PGD?
4. Scalability to Large Datasets: The paper reports a time complexity of O(n*k). While the proposed algorithm demonstrates feasibility on smaller datasets, its performance on truly large-scale datasets (e.g., ImageNet-21K) remains unclear, as the sample-wise approach may introduce significant computational overhead. Could the authors provide:
  - Profiling results detailing the actual runtime breakdown (e.g., percentage of time spent on vulnerability calculation vs. model training)
  - Empirical or simulated training overhead analysis to demonstrate how the method would scale to larger datasets.

**Ethical Concerns:**

["NO or VERY MINOR ethics concerns only"]

**Final Justification:**

The author‘s additional experiments and clarifications are appreciated. However, some concerns—particularly regarding large-scale evaluation and robustness across attack settings—remain partially unresolved. I will keep my original score as 4.

**Limitations:**

No. See the contents in Questions.

**Quality:**

2

**Strengths And Weaknesses:**

Strengths:
- (S1) The paper addresses the critical problem of the high computational cost of adversarial training. The proposed VDAT method offers a solution that not only achieves a remarkable speedup (up to 76% efficiency improvement)  but also improves the final model's natural and adversarial accuracy.
- (S2) The paper is exceptionally well-written and clearly structured.
Weakness:
- (W1) This paper has a strong experimental evaluation. However, the study could be further strengthened by incorporating adversarial training experiments on larger-scale datasets, as the benefits of the proposed algorithm may diminish in such scenarios.
- (W2) Additional experiments and analyses would be beneficial to better understand the algorithm's performance characteristics, particularly regarding how its effectiveness varies with different attack methods for vulnerability assessment.

---

> ### Author Rebuttal · Authors · 2025-07-30
>
> We are immensely grateful for your effort in reviewing our work, as well as for your constructive suggestions and insightful questions. In the following, we will address each of your concerns point-by-point.
>
> > **W1 & Q4: Adversarial training experiments on ImageNet-21K in terms of profiling results detailing the actual runtime breakdown and empirical or simulated training overhead analysis.**
>
> Thank you for the nice suggestion. We address your concern from the following two aspects.
>
> + **Profiling results detailing the actual runtime breakdown:** Due to the limitations of computational resources, the complete training of 60 epochs on ImageNet-21K will take more than 17 days, and we are sorry for not being able to provide those results in the rebuttal period. Instead, we perform VDAT along with the FGSM-based adversarial training to train ResNet50 for 10 epochs on ImageNet-21K. As shown in the table below, the time breakdown on ImageNet-21K is similar to that on ImageNet-1K. These results demonstrate that VDAT does not significantly increase the time proportion of the vulnerability calculation when the scale of the data becomes larger.
>
> |Dataset & Epochs|Vulnerability Calculation|Model Training|
> |-|-|-|
> |ImageNet-21K, Epoch=10|17.99%|82.01%|
> |ImageNet-1K, Epoch=10|17.47%|82.53%|
> |ImageNet-1K, Epoch=60|17.78%|82.22%|
>
> + **Empirical or simulated training overhead analysis:** We compare the training cost of FGSM-PGK [1] and the proposed VDAT method on ImageNet-21K. Both methods are performed to train ResNet50 for 10 epochs. As can be seen from the table below, VDAT is more efficient than FGSM-PGK. Consequently, it is demonstrated that the proposed VDAT method is still efficient on the large-scale dataset.
>
> |Methods|Cost (GPU Hours)|
> |-|-|
> |FGSM-PGK|78.53|
> |VDAT|72.66|
>
> > **W2 & Q3: It is unclear how the effectiveness of VDAT varies with different attack methods for vulnerability assessment. Would a stronger attack (e.g., PGD) lead to a more robust final model, even if the training itself still uses FGSM or PGD?**
>
> The stronger attack for vulnerability assessment can lead to a more robust final model, but it lowers the natural accuracy and increases the training cost. Specifically, we have conducted experiments on CIFAR-10 to train ResNet18 with different training losses, i.e., FGSM and PGD. Meanwhile, the adversarial attack for vulnerability assessment are changed from the default FGSM to PGD10 and PGD20. As can be seen from the table below, using stronger attacks for vulnerability assessment can increase the adversarial accuracy. However, this will also decrease the natural accuracy and training efficiency. Considering the balance of natural accuracy, adversarial accuracy, and training cost, we choose FGSM as the default attack for vulnerability assessment.
>
> |Training Loss|Vulnerability Assessment|Natural (%)|PGD50 (%)|Cost (GPU Hours)|
> |-|-|-|-|-|
> |FGSM|PGD10|85.96|58.03|1.10|
> |FGSM|PGD20|85.96|**58.50**|1.34|
> |FGSM|FGSM|**86.32**|56.23|**1.04**|
>
> |Training Loss|Vulnerability Assessment|Natural (%)|PGD50 (%)|Cost (GPU Hours)|
> |-|-|-|-|-|
> |PGD|PGD10|82.15|58.35|3.04|
> |PGD|PGD20|81.99|**58.80**|3.75|
> |PGD|FGSM|**82.96**|56.07|**1.96**|
>
> > **Q1: Could you please explain the significant accuracy improvement observed around epoch 100 in Figure 4?**
>
> The significant accuracy improvement is caused by the decay of learning rate. Specifically, the learning rate is decayed with the factor of 0.1 at the 100-th and 105-th epochs. This practice follows the convention of adversarial training [2].
>
> > **Q2: Could you please explain why VDAT_FGSM generally performs better than VDAT_PGD on both natural accuracy and adversarial accuracy? This seems a bit counterintuitive to me, since PGD provides stronger adversarial examples.**
>
> Thank you for the insightful question. We answer your question from the following two aspects.
>
> In the case of adversarial training, we would like to argue that VDAT_FGSM performs better than VDAT_PGD on natural accuracy and weak adversarial attacks, while VDAT_FGSM is not as good as VDAT_PGD on stronger attacks. As shown in Table 1, VDAT_FGSM outperforms VDAT_PGD in terms of natural accuracy and adversarial accuracy under FGSM attack. As for the stronger adversarial attacks, i.e., C&W and AA, VDAT_PGD often performs better than VDAT_FGSM. This is mainly because VDAT_PGD provides stronger adversarial examples, and learning these adversarial examples can help the model achieve better accuracy under stronger adversarial attacks. However, it is shown that stronger adversarial examples can lead to robust overfitting of the strong adversarial perturbations [3]. As a result, the model suffers from the poor generalizability to the clean data and weak adversarial examples, thus the natural accuracy and adversarial accuracy on weak adversarial attacks become lower.
>
> In the case of robust neural architecture search (NAS) in Table 4, the phenomenon mentioned by the reviewer is mainly caused by the overfitting phenomenon of the supernet. Specifically, VDAT_PGD will cause robust overfitting more easily than VDAT_FGSM during the training process of the supernet [3]. As indicated in the previous study [4], the overfitting can make the NAS algorithm add more parameter-free operations (e.g., skip connection) to the derived architecture. As a result, the architectures searched with VDAT_PGD demonstrate lower natural accuracy and adversarial accuracy than those searched with VDAT_FGSM.
>
> **References**
>
> [1] Improving fast adversarial training with prior-guided knowledge, TPAMI 2024.
>
> [2] Boosting fast adversarial training with learnable adversarial initialization, TIP 2022.
>
> [3] Understanding robust overfitting of adversarial training and beyond, ICML 2022.
>
> [4] Understanding and robustifying differentiable architecture search, ICLR 2020.

---

> > ### Comment · Reviewer_sLg7 · 2025-08-04
> >
> > Thank you for the detailed rebuttal. The additional experiments and clarifications are appreciated. However, some concerns—particularly regarding large-scale evaluation and robustness across attack settings—remain partially unresolved. I will keep my original score.

---

### Official Review · Reviewer_MotY · 2025-06-29

**Clarity:** 3
**Significance:** 3
**Originality:** 3
**Rating:** 4
**Confidence:** 4

**Summary:**

The authors introduce vulnerable data-aware adversarial training (VDAT) to improve the efficiency and effectiveness of fast adversarial training (FAT). The authors measure the vulnerability by computing the difference of decision margin for clean and adversarial examples, and then update the clean and adversarial training dataset by using adversarial examples of vulnerable examples in the training process. Extensive experiments validate the effectiveness of the proposed approach in different tasks, datasets and models.

**Questions:**

1. What are the performance when replacing the vulnerability calculation, e.g. using the pure margin, or changing the margin calculation?
2. Can the proposed VDAT combine with other multi-step or single-step adversarial training frameworks like AWP and TRADES and further improve their performance?
3. Can the proposed method be extended to other tasks apart from image classification? If so, how to measure the vulnerability and design the data filtering approach?

**Ethical Concerns:**

["NO or VERY MINOR ethics concerns only"]

**Final Justification:**

The author addressed my questions, and I believe the novelty of this paper deserves the attention of the conference audience.

**Limitations:**

yes

**Quality:**

3

**Strengths And Weaknesses:**

Strengths:

1. The proposed approach is clear elaborated with detailed procedure for each step of adversarial training, ensuring the reproducibility.
2. The paper focus on the critical issue in adversarial training, and propose an effective measurement for vulnerability of adversarial examples.
3. Extensive experiments are conducted through various tasks, datasets and models, and consistent improvement is achieved in all the experiments, validating the effectiveness of the proposed method.

Weaknesses:

1. The effectiveness of the proposed margin-based vulnerability calculation compared with other measurement is not fully explored by experimental results.
2. The proposed VDAT seems to be a plug-and-play method in multiple adversarial training frameworks. However, there lacks experiments for combining VDAT with other frameworks like AWP, TRADES and etc.

---

> ### Author Rebuttal · Authors · 2025-07-30
>
> We sincerely thank you for the time and effort you have invested in reviewing our paper. Your comments and questions are very insightful and highly valuable for us to enhance the paper. Your comments are addressed point-by-point in the following.
>
> > **W1 & Q1: The effectiveness of the proposed margin-based vulnerability calculation compared with other measurement (e.g., pure margin) is not fully explored by experimental results.**
>
> Thank you for pointing this out. To demonstrate the effectiveness of the proposed margin-based vulnerability calculation method based on soft margin, we have conducted experiments in terms of the soft margin and pure margin. Specifically, the calculation of the pure margin is shown in Equation (2) in the paper. The benchmark datasets are CIFAR-10 and CIFAR-100, and the deep neural network for adversarial training is ResNet18. As shown in the table below, both natural accuracy and adversarial accuracy decrease when changing the soft margin to the pure margin. Consequently, the proposed margin-based vulnerability calculation method is shown to be effective.
>
> |Datasets|Measurements|Natural (%)|FGSM (%)|PGD20 (%)|PGD50 (%)|
> |-|-|-|-|-|-|
> |CIFAR-10|Pure Margin|84.53|58.61|52.30|52.16|
> |CIFAR-10|Soft Margin (VDAT)|**86.32**|**62.94**|**56.30**|**56.23**|
> |CIFAR-100|Pure Margin|58.36|37.74|27.58|27.46|
> |CIFAR-100|Soft Margin (VDAT)|**59.43**|**41.17**|**35.69**|**35.61**|
>
> > **W2 & Q2: There lacks experiments for combining VDAT with other adversarial training frameworks like AWP and TRADES.**
>
> Thank you for the insightful comment. We have integrated VDAT into AWP and TRADES to perform adversarial training. In this set of experiments, the benchmark dataset is CIFAR-10 and the deep neural network for training is ResNet18. The natural accuracy, adversarial accuracy, and training cost are reported in the table below. As can be seen, VDAT can improve the natural accuracy and adversarial accuracy of both TRADES and AWP. Meanwhile, the training cost of both TRADES and AWP is decreased owing to the integration of VDAT. In summary, VDAT is shown to be effective in improving the performance of AWP and TRADES.
>
> |Methods|Natural (%)|FGSM (%)|PGD20 (%)|PGD50 (%)|Cost (GPU Hours)|
> |-|-|-|-|-|-|
> |TRADES|81.16|60.27|52.45|52.39|6.44|
> |TRADES + VDAT|**81.43**|**61.15**|**55.55**|**55.50**|**3.27**|
> |AWP|81.97|58.47|53.98|53.91|7.15|
> |AWP + VDAT|**83.97**|**62.32**|**56.03**|**55.93**|**3.76**|
>
> > **Q3: Can the proposed method be extended to other tasks apart from image classification? If so, how to measure the vulnerability and design the data filtering approach?**
>
> Yes, it can. We have shown that the proposed method can be extended to the node classification task. In this task, the vulnerability is still measured by the difference of the soft margin. Specifically, we denote the clean graph and the perturbed graph as $x$ and $x'$, and the output logit for class $c$ as $\phi_{\theta}^{c}(x_i)$, where $\theta$ denotes the parameters of GNN, and $x_i$ denotes the $i$-th node in the graph $x$. Giving the node $x_i$ and its label $y_i$, the soft margins of $x_i$ in $x$ and $x_i'$ in $x'$ are calculated by Equations (3) and (4) based on $\phi_{\theta}^{c}(x_i)$ and $\phi_{\theta}^{c}(x_i')$. Finally, the vulnerability of node $x_i$ is determined by Equation (5) based on the soft margins calculated.
>
> For the data filtering, we calculate the probability of adversarial training for each node based on Equation (6), where the vulnerability values are determined in the previous steps. According to the probability calculated, we can divide the nodes for training into two sets, i.e., $X_{adv}$ and $X_{nat}$. The nodes in $X_{adv}$ are forward to the PGD attack to change their corresponding edges. Then, the GNN is trained based on the nodes in $X_{adv}$ and $X_{nat}$.
>
> We have conducted experiments on both Cora and CiteSeer datasets with the vanilla GCN. The PGD and Min-Max adversarial attacks are implemented based on the previous study [1] with the perturbation rate of 20%. As can be seen from the table below, the proposed VDAT method can still improve the natural accuracy and adversarial accuracy while decrease the training cost.
>
> |Datasets|Methods|Natural (%)|PGD (%)|Min-Max (%)|Cost (GPU Hours)|
> |-|-|-|-|-|-|
> |Cora|PGD Train|79.00|61.00|**15.60**|0.41|
> |Cora|PGD Train + VDAT|**79.90**|**62.70**|15.10|**0.25**|
> |CiteSeer|PGD Train|69.00|52.80|8.50|0.77|
> |CiteSeer|PGD Train + VDAT|**71.70**|**54.00**|**8.80**|**0.46**|
>
> **Reference**
>
> [1] Topology attack and defense for graph neural networks: An optimization perspective, IJCAI 2019.

---

### Official Review · Reviewer_JUYs · 2025-07-01

**Clarity:** 3
**Significance:** 3
**Originality:** 3
**Rating:** 5
**Confidence:** 4

**Summary:**

This paper proposes modifying fast adversarial training (FAT) methods by making it more probable to train on adversarial examples for vulnerable training examples, creating vulnerable data-aware adversarial training (VDAT). The vulnerability of a training example is evaluated by computing the difference between the margin between the clean training example and the decision boundary and the margin between a generated adversarial example and the decision boundary. In order to have this margin account for decision boundaries for multiple classes, the paper proposes a soft margin. The probability of training on a generated adversarial example is then computed, with adversarial examples corresponding to training examples that are more vulnerable than average being more likely to be selected for use for a training parameter update. Experiments on CIFAR-10, CIFAR-100, and ImageNet-1K show that the proposed method outperforms previous methods in both natural accuracy and robust accuracy. Additional evidence for the quality of the proposed method is given by showing that combining it with neural architecture search (NAS) also leads to performance improvement.

**Questions:**

Line 174: Should R(0,1) denote a uniform random value ranged from zero to one?

What do the authors see as the value added by Proposition 4.2?

**Ethical Concerns:**

["NO or VERY MINOR ethics concerns only"]

**Final Justification:**

The paper covers a novel approach to adversarial training that focuses on vulnerable examples. The approach is thoroughly evaluated and appears to have better results than previous methods.

**Limitations:**

The big limitation of the method appears to be that it slows down FAT. This is discussed in the paper.

**Quality:**

3

**Strengths And Weaknesses:**

Quality: This paper seems to be technically sound and the proposed methods appear to be properly applied. The claims seem to be supported by the experimental results.

Clarity: The paper is clear.

Significance: The proposed method is of interest to those that want to use adversarial training for developing more robust networks.

Originality: The proposed method is, to my knowledge, novel.

---

> ### Author Rebuttal · Authors · 2025-07-30
>
> Thank you sincerely for the recognition in our work. We appreciate your great effort and answer your questions in detail below.
>
> > **Q1: Line 174: Should $\mathcal{R}(0,1)$ denote a uniform random value ranged from zero to one?**
>
> Yes, $\mathcal{R}(0,1)$ denotes a uniform random value ranged from zero to one. The aim of adopting the uniform distribution is to ensure that the probability of adding adversarial noise is exactly $\mathcal{P}_{\theta}(x_i)$. We will add the claim in terms of the uniform random value in line 174 in our next version, to make the description clearer.
>
> > **Q2: What do the authors see as the value added by Proposition 4.2?**
>
> The value of Proposition 4.2 is that it can theoretically show the effectiveness of the proposed training loss. Specifically, Proposition 4.2 can show that training with the loss in Equation (8) can lower the upper bound of the adversarial loss. Because the lower upper bound leads to better adversarial robustness, it is theoretically demonstrated that the proposed training loss can enhancing the adversarial robustness.
>
> However, as stated by Reviewer 25Wa, the current version of the theoretical analysis is still not complete in terms of the definitions, assumptions, and proofs. To avoid misleading readers, we decide to remove the theoretical claims including Proposition 4.2 in the current version. In the future, we will give a rigorous proof regarding the linear classifiers at the first step, and then conduct theoretical analysis for the deep neural networks based on the findings in the first step. We are actively working on the first step and hope to be able to present it as a complete but separate study real soon.

---

> > ### Comment · Reviewer_JUYs · 2025-08-04
> >
> > Thank you for making those changes. Based on the original submission and the author's responses, I still think that the paper should be accepted.

---

### Decision · Program_Chairs · 2025-09-17

**Decision:**

Accept (poster)

**Comment:**

This paper introduces Vulnerable Data-Aware Adversarial Training (VDAT), an effective method for enhancing the efficiency and effectiveness of Fast Adversarial Training (FAT). The work makes a significant and valuable contribution to the adversarial robustness community. It is generally well-written, with a clearly articulated motivation and methodology. The authors have also done an excellent job addressing the reviewers' concerns. For these reasons, I believe this paper would be a valuable addition to the NeurIPS program and argue for its acceptance.